# TimeStacker: A Novel Framework with Multilevel Observation for Capturing Nonstationary Patterns in Time Series Forecasting

**Qinglong Liu** [1]  **Cong Xu** [1]  **Wenhao Jiang** [1]  **Kaixuan Wang** [1]  **Lin Ma** [1]  **Haifeng Li** [1]

## Abstract

Real-world time series inherently exhibit significant non-stationarity, posing substantial challenges for forecasting. To address this issue, this paper proposes a novel prediction framework, TimeStacker, designed to overcome the limitations of existing models in capturing the characteristics of non-stationary signals. By employing a unique stacking mechanism, TimeStacker effectively captures global signal features while thoroughly exploring local details. Furthermore, the framework integrates a frequency-based self-attention module, significantly enhancing its feature modeling capabilities. Experimental results demonstrate that TimeStacker achieves outstanding performance across multiple real-world datasets, including those from the energy, finance, and weather domains. It not only delivers superior predictive accuracy but also exhibits remarkable advantages with fewer parameters and higher computational efficiency.

## 1. Introduction

Time series forecasting, which involves inferring future trends and patterns from historical observations, is widely applied in diverse domains, including weather forecasting(Wu et al., 2023), energy scheduling(Chou & Tran, 2018), traffic management(Zhou et al., 2021), medical analysis(Čepulionis & Lukoševičiūtė, 2016), and financial economics(Cheng et al., 2022). However, the complexity of real-world systems often results in non-stationary time series(Wang et al., 2024b), which complicates accurate prediction using traditional methods and presents substantial challenges for time series forecasting.

With the rapid advancement of deep learning, numerous neural network models have been developed, exhibiting remarkable performance in time series forecasting. For example, MLP-based approaches such as DLinear(Zeng et al., 2023), SOFTS(Han et al., 2024a), SparseTSF(Lin et al., 2024) and TimeMixer(Wang et al., 2024a), and Transformer-based architectures include Crossformer(Zhang & Yan, 2023), PatchTST(Nie et al., 2022), SAMformer(Ilbert et al.), and iTransformer(Liu et al., 2023). These models have achieved state-of-the-art performance in time series forecasting due to their advanced architectural designs and innovative methodologies.

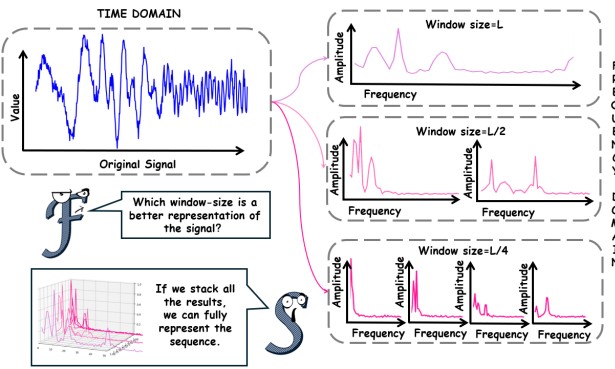

*Figure 1.* Under observations with different window sizes, the frequency of the same non-stationary signal exhibits varying patterns. The blue signal represents the original non-stationary signal, while the red components illustrate the frequency patterns obtained through Short-Time Fourier Transform (STFT) with window sizes of $L$, $2/L$, and $4/L$, respectively.

Despite the substantial advancements made by these methods in time series forecasting, the majority of studies primarily explore temporal correlations, often overlooking the frequency-domain characteristics of non-stationary signals. Based on stochastic process theory(Cox, 2017), the frequency of non-stationary signals fluctuates over time, and frequency-domain representations are more effective than time-domain signals in capturing signal characteristics within specific time intervals. Thus, analyzing the frequency variation patterns of non-stationary signals is essential for time series forecasting.

[1]Faculty of Computing, Harbin Institute of Technology, Harbin, China, Harbin, China. Correspondence to: Haifeng Li <lihaifeng@hit.edu.cn>.

*Proceedings of the $42^{nd}$ International Conference on Machine Learning*, Vancouver, Canada. PMLR 267, 2025. Copyright 2025 by the author(s).

However, the uncertainty principle of time-frequency analysis(Cohen, 1995) precludes the precise observation of a signal's frequency at a specific moment. To overcome this limitation, the short-time Fourier transform (STFT) is commonly employed to segment the original signal (i.e., divide it into patches) for frequency analysis within specific time intervals. The selection of patch size significantly influences the ability to capture frequency variation patterns: larger patches are more effective in capturing global signal features, whereas smaller patches better reveal local details. As depicted in Figure 1, signal frequencies corresponding to different patch sizes are visualized. Hence, identifying an optimal patch size for effectively extracting and representing signal patterns remains a key challenge in revealing the intrinsic regularities of sequences.

To overcome these challenges, this paper introduces a novel framework, **TimeStacker**. Rather than selecting a single optimal patch size, patches of varying sizes are sequentially stacked and aggregated based on frequency. Through iterative stacking, the most expressive patterns within the signal are progressively emphasized, enhancing the representation of the overall time series. Specifically, patterns within the signal are aggregated layer by layer in descending order of patch size, allowing the model to capture global features while retaining local details.

To further optimize the stacking process, a frequency-based enhanced self-attention module is designed to aggregate patches of the same size. Within this module, signal similarity is computed in the frequency domain, whereas aggregation operations are conducted in the time domain. This approach effectively mitigates the detrimental effects of inherent Fourier transform errors and spectral leakage in signal modeling. Experimental results indicate that TimeStacker attains state-of-the-art performance across most forecasting tasks while maintaining fewer parameters and significantly greater efficiency than other models.

The main contributions of this paper are summarized as follows:

**i)** A novel framework, **TimeStacker**, is proposed to comprehensively capture the variation patterns of frequency scales. By stacking patches sequentially from large to small, it facilitates a simple yet effective approach to time series forecasting.

**ii)** A novel frequency-based self-attention module is designed to more effectively compute the similarity between patches based on their frequencies. Additionally, this module mitigates the detrimental effects of inherent Fourier transform errors and spectral leakage in signal modeling.

**iii)** Experimentally, TimeStacker demonstrates state-of-the-art predictive accuracy across most real-world datasets while utilizing fewer parameters and exhibiting higher computa-tional efficiency than other benchmark models. This framework offers a viable solution for time series forecasting of non-stationary signals.

## 2. Related Work

Time series forecasting models have undergone substantial evolution over time. Early approaches to time series forecasting were primarily based on statistical theories, which typically assumed that time series exhibit stationarity or linear relationships(Box et al., 2015). These methods predicted trends, seasonality, and stochastic variations by modeling these components. Representative models include AutoRegressive Integrated Moving Average (ARIMA)(Lee & Tong, 2011), Exponential Smoothing (ETS)(De Livera et al., 2011), and Seasonal AutoRegressive Integrated Moving Average (SARIMA)(Dubey et al., 2021).

The emergence of deep learning facilitated significant advancements in time series forecasting, particularly with the introduction of Transformer models(Li et al., 2019). In contrast to traditional approaches, deep learning models are capable of automatically extracting features and effectively capturing complex nonlinear relationships. For example, Recurrent Neural Networks (RNNs)(Sagheer & Kotb, 2019) capture temporal dependencies in time series through their recursive structure, making them particularly effective for modeling short-term dependencies. Convolutional Neural Networks (CNNs)(Sezer et al., 2020) utilize one-dimensional convolutional operations to extract local features, effectively capturing short-term patterns and regularities in sequences.

The introduction of the Transformer architecture represented a fundamental paradigm shift in time series forecasting. For instance, PatchTST partitioned time series into independent patches embedded in high-dimensional spaces while preserving channel independence, enabling all series to share weights and establishing a foundation for subsequent research. Crossformer improved multivariate forecasting capabilities by capturing cross-dimensional patch dependencies in multivariate time series. iTransformer revisited the hierarchical design of traditional Transformer architectures, utilizing self-attention mechanisms to model inter-variable relationships and employing feedforward networks to capture nonlinear variable transformations.

Beyond Transformer-based architectures, recent years have seen substantial advancements in MLP-based models. DLinear highlighted the effectiveness of linear layers in time series forecasting, particularly excelling in long-sequence modeling. SparseTSF simplifies the forecasting task by disentangling the periodic and trend components of time series data through cross-period sparse prediction techniques. TimeMixer introduced a fully MLP-based model designed

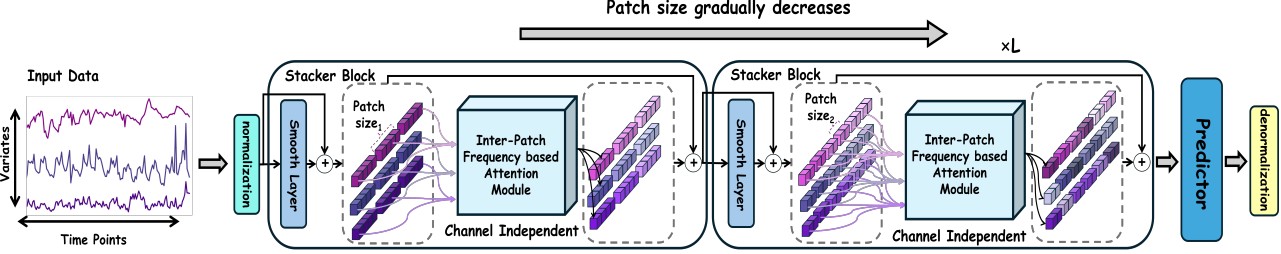

*Figure 2.* Overall Architecture of TimeStacker. The overall architecture comprises multiple Stacker Blocks. Each Stacker Block consists of a Smooth Layer and an Inter-Patch Frequency-based Attention Module, responsible for smoothing the time series and aggregating patches, respectively. Within each block, patches of the same size are aggregated based on their frequency characteristics. Subsequent blocks sequentially process patches of decreasing sizes.

to explore multiscale temporal information in time series across various temporal domains.

## 3. Method

This section begins with the task definition, followed by an overview of the preliminaries, the overall structure of TimeStacker, implementation details, theoretical analysis, and complexity analysis. The overall architecture of the proposed model is depicted in Figure 2.

### 3.1. Problem Definition

The time series forecasting problem is formulated as follows: Given a time series $X_{t-T+1:t} = \{x_{t-T+1}, \ldots, x_t\} \in \mathbb{R}^{D \times T}$, where $t$ represents a specific timestamp, $D$ is the number of variables, and $x_t \in \mathbb{R}^D$ denotes the observed value at time $t$, the objective is to predict the future values $\hat{X}_{t+1:t+\tau} = \{\hat{x}_{t+1}, \ldots, \hat{x}_{t+\tau}\} \in \mathbb{R}^{D \times \tau}$, where $\tau$ represents the prediction horizon.

### 3.2. Preliminaries

**Normalization.** Time series datasets often exhibit varying numerical ranges, which can result in unequal model attention to different data and lead to biased parameter updates. Revin(Kim et al., 2021) demonstrated that appropriate normalization significantly enhances time series forecasting performance and plays a crucial role in model training. Therefore, this model employs the same normalization and denormalization approach as Revin, utilizing mean and variance for standardization. The formulas are as follows:

$$X' = normal(X) = \frac{X - \mu}{\sigma} \qquad (1)$$

$$\hat{X} = denormal\left(\hat{X}'\right) = \hat{X}'\,\sigma + \mu \qquad (2)$$

Here, $X'$ denotes the normalized time series, $\mu$ represents the mean of the input time series, and $\sigma$ represents the standard deviation. This normalization process ensures that

the data have a mean of 0 and a variance of 1, thereby mitigating the impact of scale differences on model training.

**Channel Independence.** Channel independence is a fundamental strategy in time series forecasting. The core principle involves treating each variable in a multivariate time series as an independent channel and modeling each channel separately rather than as a whole. This approach was first introduced by PatchTST(Han et al., 2024b), which demonstrated its effectiveness in time series forecasting and has since been widely adopted in subsequent neural network models for time series prediction. This method mitigates noise interference between channels and reduces modeling complexity, thereby facilitating a more effective representation of individual variable characteristics.

### 3.3. Overall Architecture

Patches of different sizes capture the frequency variation patterns of time series to different extents. A decrease in patch size enhances the temporal resolution of the sequence while reducing its frequency resolution. Stacking patches enables a more comprehensive analysis of complex variation patterns in time series, thereby enhancing forecasting accuracy.

As illustrated in Figure 2, the overall framework comprises $L$ StackerBlocks, a normalization-denormalization module, and a predictor module. The StackerBlock is designed to capture variation patterns within patches of the same size and is elaborated in Section 3.4. The normalization-denormalization module, as discussed in Section 3.1, performs data preprocessing, while the predictor module is implemented as a linear layer.

Consider a univariate time series of length $T$, represented as $X = \{x_1, x_2, \ldots, x_T\} \in \mathbb{R}^{1 \times T}$. Given $L$ non-overlapping patch sizes defined by the vector $P = \{p_1, p_2, \ldots, p_L\}$, where $p_1 > p_2 > \ldots > p_L$ and each element in $P$ is required to be a divisor of $T$, the goal is to extract features and progressively stack patches to capture variation pat-

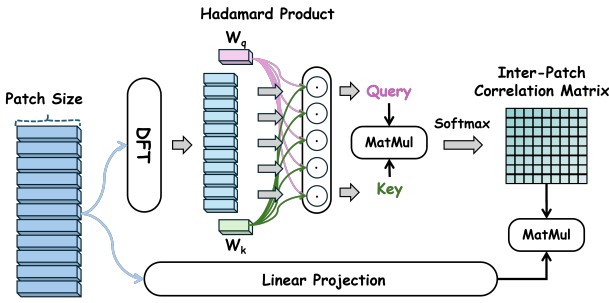

*Figure 3.* Internal Structure of FreqAttention. This module initially computes the similarity between patches in the frequency domain. It then aggregates these patches in the time domain based on the computed similarity.

terns effectively. The time series $X$ is first standardized as follows:

$$\overline{X}^{(1)} = normal(X) \qquad (3)$$

Let $p_1$ be the first element of $P$, representing the initial patch size. The standardized time series $\overline{X}^{(1)}$ is partitioned into subsequences of size $p_1$, yielding:

$$\mathcal{X}^{(1)} = \left\{ \S_1^{(1)}, \S_2^{(1)}, \ldots, \S_{k_1}^{(1)} \right\} \in \mathbb{R}^{k_1 \times p_1} \qquad (4)$$

Here $k_1 = \frac{T}{p_1}$. Next, $\S_1^{(1)}$ is fed into the StackerBlock associated with $p_1$, which extracts variation patterns within patches of the same size, yielding $\overline{\mathcal{X}}^{(1)}$. Similarly, let $p_2$ be the next patch size, and re-segment $\overline{\mathcal{X}}^{(1)}$ into patches, yielding:

$$\mathcal{X}^{(2)} \in \mathbb{R}^{k_2 \times p_2}, k_2 = \frac{T}{p_2} \qquad (5)$$

After applying the same operations, $\overline{\mathcal{X}}^{(2)}$ is obtained. This process is iterated $L$ times, with each step further partitioning patches based on the preceding iteration. Through iterative patch stacking, the variation patterns of the time series are progressively captured by integrating patches of different sizes. The formal representation is as follows:

$$\mathcal{X}^{(l)} = Concat \begin{bmatrix} StackerBlock_{l-1}\left(\mathcal{X}^{(l-1)}\right)_1 \\ StackerBlock_{l-1}\left(\mathcal{X}^{(l-1)}\right)_2 \\ StackerBlock_{l-1}\left(\mathcal{X}^{(l-1)}\right)_3 \\ \ldots \\ StackerBlock_{l-1}\left(\mathcal{X}^{(l-1)}\right)_{p_l} \end{bmatrix}, \qquad (6)$$
$$l = 1, 2, \ldots, L$$

### 3.4. Stacker Block

To effectively capture variation patterns across patches (inter-patch), the **StackerBlock** is introduced. This module

comprises two core components: the **Smooth Layer** and the **Inter-Patch Frequency-Based Attention Module**.

In real-world time series, outliers can significantly impact the model's ability to capture sequence variation patterns. To mitigate this issue, the Smooth Layer is introduced, utilizing time points within a patch to reduce the influence of outliers. Specifically, the Smooth Layer is implemented using a convolution operation with a kernel size of $p_l$. The formal representation is given by:

$$SmoothLayer_l\left(\mathcal{X}^{(l)}\right) = W * \mathcal{X}^{(l)} + b \qquad (7)$$

To exploit frequency variation information within the sequence, the **Inter-Patch Frequency-Based Attention Module** (**FreqAttention**) is introduced. The structure of FreqAttention is depicted in Figure 3. Unlike traditional self-attention mechanisms, similarity in this approach is computed in the frequency domain. The formal representation is given by:

$$\widetilde{\mathcal{X}}^{(l)} = \mathcal{F}\left(\mathcal{X}^{(l)}\right) \qquad (8)$$

$$Q = W_q \odot \widetilde{\mathcal{X}}^{(l)}, \ K = W_k \odot \widetilde{\mathcal{X}}^{(l)} \qquad (9)$$

$$CorMat = Softmax\left(\frac{W_q \odot \widetilde{\mathcal{X}}^{(l)} \cdot (W_k \odot \widetilde{\mathcal{X}}^{(l)})^T}{\sqrt{d_k}}\right) \qquad (10)$$

Here, $\mathcal{F}(\cdot)$ denotes the Fourier transform, $\odot$ represents the Hadamard product, and $W_q, W_k \in \mathbb{R}^{\lfloor \frac{p_l}{2} \rfloor + 1}$ are learnable parameters used to compute the query and key vectors, and $d_k$ is a scalar matching the dimension of $widetilde \mathcal{X}^{(l)}$. From a signal processing perspective, the Hadamard product enables $W_q$ and $W_k$ to function as learnable filters, extracting frequency components relevant to subsequent sequences. Consequently, $CorMat \in \mathbb{R}^{k_l \times k_l}$ represents the frequency-based similarity between patches and reflects the sequence's variation patterns in the frequency domain. Aggregating the time series in the time domain using $CorMat$ yields the following formal expression:

$$V = \mathcal{X}^{(l)} W_v \qquad (11)$$

$$FreqAttn(\mathcal{X}^{(l)}) = \\ Softmax\left(\frac{W_q \odot \widetilde{\mathcal{X}}^{(l)} \cdot (W_k \odot \widetilde{\mathcal{X}}^{(l)})^T}{\sqrt{d_k}}\right) \mathcal{X}^{(l)} W_v \qquad (12)$$

Here, $W_v \in \mathbb{R}^{p_l}$ is a learnable parameter for computing the value vector. By combining Equations (7) to (12), the formal expression of the StackerBlock is given by:

$$StackerBlock_l(\mathcal{X}^{(l)}) = \\ FreqAttn(SmoothLayer_l(\mathcal{X}^{(l)}) + \mathcal{X}^{(l)}) + \mathcal{X}^{(l)} \qquad (13)$$

*Table 1.* Main Results. All results are based on input sequences of length 96 and are calculated as the average across four different prediction lengths {96, 192, 336, 720}. The prediction performance is evaluated using MSE or MAE as metrics, where lower values indicate closer alignment between the predicted and actual sequences. Complete experimental results are provided in Appendix D.1.

| Models | TimeStacker (ours) | | SOFTS (2024) | | SparseTSF (2024) | | iTransformer (2024) | | TimeMixer (2024) | | SAMformer (2024) | | PatchTST (2023) | | Crossformer (2023) | | DLinear (2023) | | RLinear (2023) | |
|---|---|---|---|---|---|---|---|---|---|---|---|---|---|---|---|---|---|---|---|---|---|
| Metric | MSE | MAE | MSE | MAE | MSE | MAE | MSE | MAE | MSE | MAE | MSE | MAE | MSE | MAE | MSE | MAE | MSE | MAE | MSE | MAE |
| ETTh1 | **0.433** | **0.423** | 0.449 | 0.442 | _0.441_ | _0.425_ | 0.454 | 0.447 | 0.447 | 0.440 | 0.444 | 0.432 | 0.469 | 0.454 | 0.529 | 0.522 | 0.456 | 0.452 | 0.446 | 0.434 |
| ETTh2 | _0.368_ | **0.390** | 0.373 | 0.400 | 0.421 | 0.438 | 0.383 | 0.407 | **0.364** | _0.395_ | 0.383 | 0.401 | 0.387 | 0.407 | 0.942 | 0.684 | 0.559 | 0.515 | 0.374 | 0.398 |
| ETTm1 | **0.381** | **0.381** | 0.393 | 0.403 | 0.425 | 0.401 | 0.407 | 0.410 | **0.381** | _0.395_ | 0.415 | 0.407 | 0.387 | 0.400 | 0.513 | 0.496 | 0.403 | 0.407 | 0.414 | 0.407 |
| ETTm2 | **0.274** | **0.316** | 0.287 | 0.330 | 0.297 | 0.331 | 0.288 | 0.332 | _0.275_ | _0.323_ | 0.285 | 0.327 | 0.281 | 0.326 | 0.757 | 0.610 | 0.350 | 0.401 | 0.286 | 0.327 |
| Traffic | 0.508 | 0.335 | **0.409** | **0.267** | 0.578 | 0.350 | _0.428_ | _0.282_ | 0.484 | 0.297 | 0.595 | 0.382 | 0.481 | 0.304 | 0.550 | 0.304 | 0.625 | 0.383 | 0.626 | 0.378 |
| Electricity | 0.194 | 0.275 | **0.174** | **0.264** | 0.222 | 0.289 | _0.178_ | _0.270_ | 0.182 | 0.272 | 0.217 | 0.295 | 0.205 | 0.290 | 0.244 | 0.334 | 0.212 | 0.300 | 0.219 | 0.298 |
| Weather | _0.243_ | **0.264** | 0.255 | 0.278 | 0.290 | 0.302 | 0.258 | 0.278 | **0.240** | _0.271_ | 0.264 | 0.285 | 0.259 | 0.348 | 0.259 | 0.315 | 0.265 | 0.317 | 0.272 | 0.291 |
| Exchange | **0.336** | **0.389** | 0.348 | _0.395_ | 0.365 | 0.401 | 0.360 | 0.403 | 0.355 | 0.399 | _0.346_ | 0.399 | 0.367 | 0.404 | 0.940 | 0.707 | 0.354 | 0.414 | 0.378 | 0.417 |

## 3.5. Theoretical Analysis

**Definition 3.1.** The statistical properties of non-stationary signals (e.g., mean, variance, autocorrelation function) change over time, and their frequency characteristics also dynamically evolve with time. These can be expressed using the Fourier series as follows:

$$x(t) = a_0(t) + \sum_{n=1}^{\infty}(a_n(t)\cos(2\pi n f_0 t) + b_n(t)\sin(2\pi n f_0 t)) \tag{14}$$

Let $x(t)$ denote a non-stationary signal, where $a_n(t)$ and $b_n(t)$ are time-varying nonlinear functions that capture the dynamic variations in the signal components. This implies that the frequency content of a non-stationary signal varies over time.

From a frequency-domain perspective, time series forecasting fundamentally involves analyzing the latent frequency characteristics within historical sequences to identify the temporal variation patterns of their Fourier coefficients $a_n(t)$ and $b_n(t)$. These coefficients encode the amplitude and phase information of the signal at specific frequencies, serving as key indicators of its time-varying properties in the frequency domain. Therefore, capturing and modeling the variation patterns of these coefficients can effectively reveal the periodicity and trends in time series, establishing a robust foundation for accurate future forecasting.

**Theorem 3.2.** *The time-frequency uncertainty principle states that a signal cannot achieve arbitrarily high resolution simultaneously in both the time and frequency domains, reflecting the resolution limit of a signal in the time-frequency domain. The mathematical expression is as fol-*

*lows:*

$$\Delta t \cdot \Delta f \geq \frac{1}{4\pi} \tag{15}$$

Here $\Delta t$ denotes the standard deviation in the time domain, while, and $\Delta f$ denotes the standard deviation in the frequency domain. This reveals an inherent limitation in precisely measuring both the temporal location and frequency components of a signal: achieving arbitrarily high time and frequency resolution simultaneously is fundamentally impossible.

To overcome this limitation, TimeStacker sequentially stacks patches of varying sizes, from large to small, effectively reducing $\Delta f$ (frequency resolution) while progressively enhancing time resolution. This approach enables the model to capture the dynamic evolution of the spectrum over time at multiple temporal resolutions, allowing it to focus on both fine-grained local details and overarching global trends.

Specifically, larger patches yield lower time resolution and higher frequency resolution, enabling the model to capture global periodicity and long-term trends within the signal. Conversely, smaller patches yield higher time resolution, allowing the model to accurately capture dynamic variations in local signal components.

## 3.6. Complexity Analysis

The Inter-Patch Frequency-based Attention Module distinguishes itself from traditional self-attention mechanisms primarily in the computation of $Q$ and $K$. To facilitate explanation, we use the calculation of $Q$ as an example for analyzing computational complexity.

Let the input for similarity computation be $\mathcal{X} \in \mathbb{R}^{k \times p}$.

*Table 2.* Ablation Experiment Results. Ablation experiments were conducted on the FreqAttention module of TimeStacker, involving replacement(REPLACE) and removal(W/O) operations based on traditional self-attention. MAE and MSE were used as evaluation metrics, with all input sequence lengths set to 96.

| Method | | W/O FreqAttention | REPLACE FreqAttention | W/O Hadamard | REPLACE Hadamard | TimeStacker |
|---|---|---|---|---|---|---|
| | | MSE MAE | MSE MAE | MSE MAE | MSE MAE | MSE MAE |
| ETTh1 | 96 | 0.388 0.395 | 0.386 0.395 | 0.384 0.393 | 0.381 0.393 | **0.379 0.385** |
| | 192 | 0.436 0.420 | 0.431 0.424 | 0.435 0.425 | 0.431 0.424 | **0.429 0.416** |
| | 336 | 0.476 0.444 | 0.472 0.445 | 0.472 0.444 | 0.474 0.447 | **0.459 0.436** |
| | 720 | 0.480 0.471 | 0.474 0.469 | 0.466 0.464 | 0.476 0.468 | **0.464 0.455** |
| ETTh2 | 96 | 0.288 0.338 | 0.296 0.343 | 0.292 0.338 | 0.293 0.342 | **0.280 0.327** |
| | 192 | 0.398 0.397 | 0.389 0.396 | 0.390 0.399 | 0.393 0.394 | **0.373 0.385** |
| | 336 | 0.418 0.426 | 0.416 0.424 | 0.419 0.426 | 0.414 0.423 | **0.407 0.416** |
| | 720 | 0.418 0.437 | 0.419 0.437 | 0.420 0.439 | 0.418 0.436 | **0.412 0.431** |
| Exchange | 96 | 0.087 0.206 | 0.085 0.203 | 0.086 0.204 | 0.085 0.203 | **0.084 0.200** |
| | 192 | 0.179 0.303 | 0.173 0.293 | 0.171 0.294 | 0.171 0.293 | **0.171 0.293** |
| | 336 | 0.323 0.415 | 0.317 0.411 | 0.320 0.412 | 0.316 0.410 | **0.314 0.408** |
| | 720 | 0.807 0.675 | 0.791 0.669 | 0.780 0.666 | 0.796 0.672 | **0.776 0.656** |

The computation of $Q$ is detailed in Equations (8) and (9). The computational complexity of the Fast Fourier Transform (FFT) is $O\left(kplog_2p\right)$, while the complexity of the Hadamard product is $O\left(k\left(\left\lfloor\frac{p}{2}\right\rfloor+1\right)\right)$ approximately $O\left(k\frac{p}{2}\right)$. Substituting $L = kp$ into the formulas, the complexity of computing $Q$ is derived as $O\left(L\left(log_2p+\frac{1}{2}\right)\right)$, which approximates to $O\left(Llog_2p\right)$. In comparison, the computational complexity of traditional self-attention is $O\left(kp^2\right)$, which simplifies to $O\left(Lp\right)$. This demonstrates that the proposed method significantly reduces complexity, particularly for high-dimensional inputs, where its advantages become especially evident.

This method leverages the frequency characteristics of signals to alleviate the performance bottlenecks in traditional self-attention models, which are caused by high computational complexity in the time domain. This optimization provides a more efficient computational pathway for time series forecasting, maintaining the model's predictive accuracy and robustness.

# 4. Experiments

In this section, extensive experiments and analyses are conducted on real-world datasets from various domains, encompassing both long-term and short-term forecasting tasks, to comprehensively evaluate the performance and computational efficiency of TimeStacker.

## 4.1. Experimental Setup

**Dataset.** To evaluate the performance of TimeStacker, experiments were conducted on multiple widely used real-world datasets, following a processing approach similar to

iTransformer(Liu et al., 2023). These datasets span various domains, including energy, transportation, and weather. Specifically, the datasets used in the experiments include ETT (comprising four subsets: ETTh1, ETTh2, ETTm1, and ETTm2), Weather, Traffic, Electricity, and Exchange. Each dataset was divided into training, validation, and test sets in a 6:2:2 ratio. For more details on the datasets, refer to Appendix A.

**Implementation Details**. TimeStacker employs Mean Absolute Error (MAE) as the loss function and utilizes Adam as the optimizer. The learning rate is set to $1 \times 10^{-3}$, the weight decay to $1 \times 10^{-3}$, and $\epsilon$ to $1 \times 10^{-8}$. The first and second moment decay rates are set to 0.9 and 0.999, respectively. All experiments were implemented using PyTorch 2.0(Paszke et al., 2019) and conducted on an NVIDIA RTX 4080 GPU with 16GB of memory. Detailed parameter settings can be found in Appendix B.2.

## 4.2. Experimental Results

**Baseline.** We compared TimeStacker with state-of-the-art and representative models proposed in the past two years to evaluate its effectiveness. The main baselines include: (1) Transformer-based models such as Pathformer(Chen et al., 2024), SAMformer(Ilbert et al.), PatchTST(Nie et al., 2022), and Crossformer(Zhang & Yan, 2023); (2) Linear-layer-based models such as SparseTSF(Lin et al., 2024), SOFTS(Han et al., 2024a), TimeMixer(Wang et al., 2024a), DLinear(Zeng et al., 2023), and RLinear(Li et al., 2023).

**Main Results.** Table 1 presents the time series forecasting results. Red highlights the best performance, while blue with underlining indicates the second-best performance. Lower Mean Squared Error (MSE) and MAE values cor-

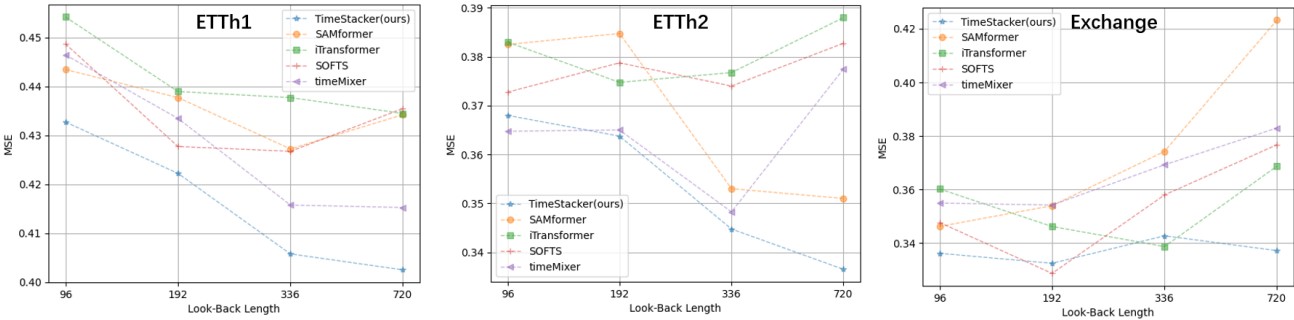

*Figure 4.* Impact of Look-Back Lengths. All models use look-back window lengths of 96, 192, 336, and 720, with MSE serving as the evaluation metric. The results are calculated as the average across four different prediction lengths {96, 192, 336, 720}, to assess the performance of different models under varying look-back lengths.

*Table 3.* The impact of different look-back window lengths on model prediction performance. The look-back window lengths are {96, 192, 336, 720}, and the prediction lengths are {96, 192, 336, 720}. The evaluation metric is MSE.

| Method | | 96 | 192 | 336 | 720 |
|---|---|---|---|---|---|
| | | MSE | MSE | MSE | MSE |
| ETTh1 | 96 | 0.379 | 0.370 | **0.362** | 0.364 |
| | 192 | 0.429 | 0.424 | 0.401 | **0.399** |
| | 336 | 0.459 | 0.448 | 0.428 | **0.42** |
| | 720 | 0.464 | 0.447 | 0.432 | **0.427** |
| ETTh2 | 96 | 0.280 | 0.283 | 0.270 | **0.267** |
| | 192 | 0.373 | 0.366 | 0.348 | **0.336** |
| | 336 | 0.407 | 0.397 | 0.362 | **0.354** |
| | 720 | 0.412 | 0.409 | 0.399 | **0.389** |
| Exchange | 96 | **0.084** | **0.084** | 0.085 | 0.085 |
| | 192 | **0.171** | 0.173 | 0.174 | **0.171** |
| | 336 | **0.314** | 0.317 | 0.320 | 0.321 |
| | 720 | 0.776 | **0.756** | 0.792 | 0.772 |

respond to higher predictive accuracy. Compared to other models, TimeStacker demonstrates superior performance across multiple datasets (ETTh1∼ETTm2, Weather, Exchange). Notably, in comparison with iTransformer, TimeStacker achieves significant improvements. For example, on the ETTm1 dataset, MAE is reduced by 7.07%; on the Weather dataset, MAE is reduced by 5.04%; and on the Exchange dataset, MSE is reduced by 6.67%. These results clearly demonstrate that TimeStacker not only excels in time series forecasting accuracy but also exhibits strong generalization capabilities. In particular, in critical application domains such as energy, weather, and finance, TimeStacker showcases broad applicability and practical utility.

### 4.3. Model Analysis

**Ablation Study.** A series of ablation experiments were conducted to evaluate the core components of TimeStacker, namely the FreqAttention module. These experiments in-

volved replacement (REPLACE) and removal (W/O) operations, as outlined below:

- *W/O FreqAttention*: Removed the FreqAttention module from TimeStacker.

- *REPLACE FreqAttention*: Replaced the FreqAttention module in TimeStacker with traditional Self-Attention.

- *W/O Hadamard*: Removed the Hadamard product operation from the FreqAttention module.

- *REPLACE Hadamard*: Replaced the Hadamard product operation in the FreqAttention module with a linear transformation.

As shown in Table 2, incorporating frequency information into similarity computation significantly enhances the accuracy of time series forecasting compared to traditional self-attention methods. Introducing the Hadamard product in the computation of $Q$ and $K$ further reveals hidden pattern features within the sequence. The proposed FreqAttention exhibits a distinct advantage in capturing time series dynamics, confirming its effectiveness in modeling the complex characteristics of dynamic signals.

**Impact of Look-Back Lengths.** In general, longer lookback windows provide more information about the sequence, allowing the model to capture sequence features more comprehensively. However, longer sequences may also introduce additional noise, potentially affecting prediction accuracy. To examine the impact of look-back window length on model performance, experiments were conducted using varying window lengths of {96, 192, 336, 720}.

The detailed results, presented in Table 3, indicate that prediction accuracy on the Exchange dataset remained largely unchanged regardless of window length. In contrast, for the ETTh1 and ETTh2 datasets, accuracy significantly improved as the look-back window length increased. These

findings suggest that TimeStacker effectively utilizes the additional sequence information from longer windows to enhance forecasting performance.

Additionally, our model was compared with SOFT, TimeMixer, iTransformer, and SAMformer. As illustrated in Figure 4, the results demonstrate the superior ability of TimeStacker in capturing sequence features. Further details can be found in Appendix D.2.

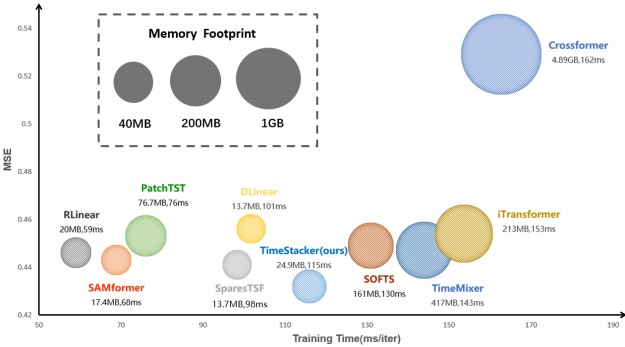

*Figure 5.* Performance Comparison of Models. All models were evaluated on the ETTh1 dataset using an experimental setup with a look-back length of 96, a prediction length of 720, and a batch size of 128. The final evaluation results were obtained by averaging the outcomes of five independent trials. The evaluation metrics included memory usage during runtime, execution time, and MSE.

**Model Effectiveness.** In Section 3.6, the computational complexity of the core module in TimeStacker was derived as $O\left(Llog_2p\right)$. To provide a more comprehensive evaluation of the model's efficiency, additional analyses were conducted on memory usage and training time. The ETTh1 dataset was used for this evaluation, with all models configured to a look-back length of 96, a prediction length of 720, and a batch size of 128. The results were averaged over five independent trials.

As illustrated in Figure 5, the models with the lowest memory usage were SparseTSF and DLinear, each consuming only 13.7 MB, while the fastest model was RLinear, with an iteration time of just 59 ms. In comparison, TimeStacker required 24.9 MB of memory and had an iteration time of 115 ms, while achieving the lowest MSE among all models. Clearly, TimeStacker demonstrates strong predictive performance and computational efficiency, achieving the lowest prediction error while maintaining relatively low memory consumption and runtime.

## 5. Conclusion, Limitation, and Future Works

**Conclusion.** This paper introduces the TimeStacker framework, which leverages the frequency variations of nonstationary signals over time. By employing a unique stack-

ing mechanism and balancing modeling between the time and frequency domains, TimeStacker effectively captures the complex characteristics of non-stationary time series. To further enhance the stacking mechanism and fully utilize frequency information, a frequency-based self-attention module was designed. This module not only mitigates the impact of spectral leakage but also enhances feature representation by comprehensively modeling both frequency and time domain information.

Additionally, extensive experiments were conducted on various real-world datasets. The results demonstrate that TimeStacker effectively extracts the dynamic characteristics of non-stationary time series and achieves a comprehensive feature representation. Performance analysis further reveals that this approach delivers high prediction accuracy while significantly reducing computational complexity, highlighting its potential for applications in resource-constrained scenarios.

**Limitation and Future Work.** While TimeStacker effectively captures time series features, comparative experiments indicate a slight decline in predictive performance as the number of variables in the time series increases. To investigate this phenomenon, additional experiments were conducted (in Appendix D.5) by varying the number of variables and testing the model on the Traffic and Electricity datasets. The results suggest that the model may encounter performance bottlenecks when handling multivariate time series.

Therefore, future work will focus on optimizing multi-channel prediction strategies by designing additional modules to enhance TimeStacker's performance on multivariate time series tasks.

## Acknowledgements

This study is supported by the Young Scientists Fund of the National Natural Science Foundation of China(Grant No.62306094), Independent Research Exploration Projects of Songjiang Laboratory(Grant No.SL20230203), Project supported by the Special Funds of the National Natural Science Foundation of China(Grant No. 32441112).

## Impact Statement

This paper presents work whose goal is to advance the field of Machine Learning. There are many potential societal consequences of our work, none which we feel must be specifically highlighted here.

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

## A. DATASET DESCRIPTION

Time series forecasting experiments were conducted on widely used real-world datasets (details provided in Table 4). The datasets are described as follows:

- ETT Dataset: Comprising two hourly datasets (ETTh) and two 15-minute datasets (ETTm), this dataset includes load features of seven oil and electricity transformers recorded from July 2016 to July 2018.

- Traffic Dataset: This dataset contains hourly road occupancy rates recorded by sensors on San Francisco freeways from 2015 to 2016.

- Electricity Dataset: Recording hourly electricity consumption for 321 customers, this dataset spans the period from 2012 to 2014.

- Weather Dataset: Comprising 21 weather indicators, including air temperature and humidity, this dataset was recorded every 10 minutes throughout 2020.

- Exchange-rate Dataset: This dataset contains daily exchange rates for eight countries, collected from 1990 to 2016.

*Table 4. Variates* denotes the number of variables in each dataset, *Dataset Size* denotes the number of time points in the dataset, Frequency denotes the sampling frequency of the dataset, and Information denotes the category information of the dataset.

| Dataset | ETTh1 | ETTh2 | ETTm1 | ETTm2 | Traffic | Electricity | Weather | Exchange-rate |
|---|---|---|---|---|---|---|---|---|
| **Variates** | 7 | 7 | 7 | 7 | 862 | 321 | 21 | 8 |
| **Dataset Size** | 14,307 | 14,307 | 57,507 | 57,507 | 17,451 | 26,211 | 52,603 | 7,207 |
| **Frequency** | Hourly | Hourly | 15min | 15min | Hourly | Hourly | 10min | Daily |
| **Information** | Electricity | Electricity | Electricity | Electricity | Transportation | Electricity | Weather | Economy |

## B. IMPLEMENT DETAILS

### B.1. TimeStacker

The complete algorithmic process of TimeStacker is outlined as follows. It takes the time series $x$ as input and generates the corresponding prediction $\widetilde{x}$.

---
**Algorithm 1** TimeStacker

---
    **Input:** Historical look-back window $x_{t-T+1:t} \in \mathbb{R}^T$, Patch size list $P = \{p_1, p_2, \ldots, p_L\}$
    **Output:** Forecasting horizon $\widetilde{x}_{t+1:t+H} \in \mathbb{R}^H$,
    $\bar{x} = Normalize(x)$ /* Normalizer the input sequence with mean and variance */
    **for** $p_i$ **in** $P$ **do**
        $\bar{x} = SmoothLayer(\bar{x})$ /* Apply $conv1d$ with kernel size of $p_i$ */
        $\bar{x} = Reshape(\bar{x}, (p_i, T/p_i))$
        $\bar{x} = FreqAttention(\bar{x})$ /* Aggregate different patches */
    **end for**
    $\bar{x} = Reshape(\bar{x}, T)$
    $\widetilde{x} = Denormalize(predictor(\bar{x}))$ /* Apply linear layer for prediction */

---

### B.2. Model Configuration

TimeStacker primarily consists of multiple Stacker Blocks, with the number of blocks determined by the Patchlist parameter. The number of elements in Patchlist defines the depth of TimeStacker, while the size of each element determines the observation window for the corresponding block. Given the varying sampling frequencies and physical characteristics of different datasets, customized configurations are adopted for each dataset. Detailed information is provided in Table 5.

*Table 5.* Model Configuration.

| DataSet | ETTh1 | ETTh2 | ETTm1 | ETTm2 | Traffic | Electricity | Weather | Exchange |
|---|---|---|---|---|---|---|---|---|
| Patch Size List | (96,48,32,24,16,12) | (96,48,32,24,16,12) | (48,32,24,16) | (48,32,24,16) | (96,48,32,16,12) | (96,48,24,12) | (96,48,32,24,12) | (96,48,24) |

## C. EXPERIMENT DETAILS

### C.1. Metric Details

To comprehensively evaluate model performance across different datasets, Mean Absolute Error (MAE) and Mean Squared Error (MSE) were selected as evaluation metrics. These metrics assess prediction accuracy from different perspectives, offering a comprehensive and intuitive basis for model comparison.

**MAE:**

$$MAE = \frac{1}{L}\sum_{l=1}^{L}\left|x_i - \widetilde{x}_i\right| \tag{16}$$

**MSE:**

$$MSE = \frac{1}{L}\sum_{l=1}^{L}\left(x_i - \widetilde{x}_i\right)^2 \tag{17}$$

Here, $x_i$, $\widetilde{x}_i \in \mathbb{R}^{C \times L}$ denote the ground truth and predicted values, respectively, where $L$ represents the number of time points, and $C$ is the number of channels. The MSE primarily emphasizes the squared differences between predicted and actual values, thereby amplifying the impact of larger errors. This makes MSE particularly sensitive to outliers and suitable for evaluating the model's ability to capture global trends. On the other hand, the MAE directly computes the average absolute differences between predicted and actual values. It focuses on assessing the model's overall control of error magnitude in practical applications and is less affected by outliers.

### C.2. Baseline

Representative methods from the past two years in time series forecasting were selected as baseline approaches to comprehensively evaluate the performance of the proposed model. A detailed introduction to these methods is provided below:

**DLinear:** Utilizes a simple yet effective single-layer linear model to capture temporal relationships between input and output sequences.

**Crossformer:** Segments multivariate time series data along each dimension, embeds them into feature vectors, and employs a two-stage attention mechanism to efficiently capture both intra- and inter-series dependencies.

**PatchTST:** Splits time series data into subsequence-level patches to extract local semantics, adopting a channel-independent strategy where each channel shares the same embedding and Transformer weights across all sequences.

**RLinear:** Uses linear mapping to model periodic features in multivariate time series, demonstrating robustness across different periods as input length increases.

**SAMformer:** Enhances the model's generalization ability by leveraging sharpness-aware optimization techniques.

**TimeMixer:** Addresses complex temporal variations in time series forecasting through a multi-scale mixing perspective, improving complementary predictions from multi-scale sequences by decoupling variations.

**iTransformer:** Reverses the Transformer structure by encoding each individual series as variable tokens without modifying any existing modules.

**SparseTSF:** Simplifies the forecasting task by decoupling the periodicity and trend of time series data using cross-period sparse prediction techniques.

**SOFTS:** Introduces a novel centralized structure to transfer information across channels, addressing the limitations of

channel-independent approaches in leveraging inter-channel correlations and mitigating robustness challenges in channel-dependent methods.

### C.3. Experiment details

TimeStacker adopts MAE as the loss function and uses Adam as the optimizer, with a learning rate set to $1 \times 10^{-3}$, weight decay set to $1 \times 10^{-3}$, and epsilon set to $1 \times 10^{-8}$. The first-order and second-order moments are configured as 0.9 and 0.999, respectively. All experiments are implemented in PyTorch 2.0 and executed on an NVIDIA RTX 4080 GPU with 16GB of memory.

In the experiments, the same dataset processing approach as TimeMixer was followed, ensuring that datasets were divided into training, validation, and test sets in a strict temporal sequence with a 6:2:2 ratio to prevent data leakage. The look-back window length was fixed at 96, and the forecasting horizons were set to {96,192,336,720}.

## D. FULL RESULTS

### D.1. Complete Experimental Result

The complete experimental results are presented in Table 12. Experiments were conducted on six widely used real-world datasets spanning domains such as energy, traffic, and weather to comprehensively validate the effectiveness of the proposed method. To further assess model performance, comparisons were made against several representative models in the field.

The results demonstrate that the proposed method achieves outstanding performance across multiple datasets, consistently surpassing most baseline models in both predictive accuracy and computational efficiency. Notably, the method exhibits significant advantages in handling complex non-stationary time series, effectively capturing both global trends and local details within the signals. These findings further underscore the generalizability and robustness of the approach, offering an efficient and accurate solution for time series forecasting tasks.

The results of the mean and standard deviation of MSE and MAE for multiple runs of all datasets are shown in Table 6.

*Table 6.* Mean/Standard deviation for MSE and MAE across multiple runs.

| DataSet | ETTh1 | ETTh2 | ETTm1 | ETTm2 | Traffic | Electricity | Weather | Exchange |
|---------|-------|-------|-------|-------|---------|-------------|---------|----------|
| MSE | 0.433/0.00145 | 0.368/0.00091 | 0.381/0.00119 | 0.274/0.00061 | 0.508/0.00052 | 0.194/0.00056 | 0.243/0.00092 | 0.336/0.00101 |
| MAE | 0.423/0.00167 | 0.390/0.00057 | 0.381/0.00052 | 0.316/0.00042 | 0.335/0.00087 | 0.275/0.00077 | 0.264/0.00042 | 0.389/0.00137 |

### D.2. Impact of Look-Back Lengths

The impact of look-back length on model performance was examined, with detailed experimental results presented in Table 11. Prediction experiments were conducted using look-back lengths of {96, 192, 336, 720}, and the results were compared against SOFT, TimeMixer, iTransformer, and SAMformer. The results demonstrate that as the look-back length increases, the predictive accuracy of TimeStacker improves significantly. Compared to other models, TimeStacker effectively utilizes the extended sequence information, resulting in a substantial enhancement in prediction performance. This outcome validates the advantages and robustness of the proposed method in time series modeling.

### D.3. Impact of Patch Size List

To demonstrate how TimeStacker adapts to various non-stationary signals, we configured the parameter Patch Size List and conducted experiments on the ETTm1 dataset. The results are shown in Table 7. These results indicate that employing various window combinations can more effectively capture the underlying dynamic patterns of the sequence, thereby improving prediction performance.

### D.4. Complexity Analysis

To enable a more in-depth analysis of the model's computational complexity, the input length was increased to evaluate its temporal and spatial complexity (GPU Memory (MB) / Training Time (ms/iter)). The experimental results are shown in

*Table 7.* Impact of different patch size list.

| Patch Size List | [16, 16, 16, 16] | [16, 16, 16, 24] | [16, 16, 16, 32] | [16, 16, 16, 48] | [16, 16, 24, 32] | [16, 16, 24, 48] | [16, 24, 32, 48] |
|---|---|---|---|---|---|---|---|
| MSE | 0.465 | 0.468 | 0.468 | 0.465 | 0.463 | 0.463 | 0.460 |
| MAE | 0.433 | 0.439 | 0.436 | 0.431 | 0.431 | 0.430 | 0.428 |

*Table 8.* Complexity Analysis(GPU Memory(MB)/Training Time(ms/iter)).

| Input Length | Timestacker | SparseTSF | TimeMixer | DLinear | PatchTST | Crossformer |
|---|---|---|---|---|---|---|
| 192 | 28.8/134 | 15.6/108 | 518.3/180 | 13.6/45 | 145.8/87.8 | 5214/238 |
| 384 | 29.3/133 | 16.1/112 | 875.4/193 | 16.7/49 | 334.7/90.1 | 5734/273 |
| 768 | 34.3/137 | 20.6/115 | 1763/202 | 23.2/99 | 830.0/93.3 | 6814/342 |
| 1536 | 59.2/137 | 28.7/127 | 3744/282 | 34.8/108 | 2404/137 | 9016/1007 |
| 3072 | 110.7/134 | 44.1/131 | 7376/462 | 59.1/108 | 7832/1315 | 12470/3121 |

Table 8.

### D.5. Additional Experiments

To investigate the impact of the number of variables on model performance, we selected the Traffic and Electricity datasets, where our model performed less effectively, and conducted comparative experiments with existing models including SOFTS, iTransformer, and TimeMixer. These experiments focused specifically on single-variable time series forecasting tasks.

The results not only revealed the differences in how various models handle single-channel time series but also provided a clearer understanding of how the number of variables affects model performance. Under single-channel conditions, our model demonstrated a stronger ability to capture the dynamic changes of individual sequences, whereas certain other models appeared to rely on correlations between multiple channels to enhance predictive accuracy. Furthermore, these findings offer valuable insights for optimizing model design, such as exploring strategies to balance feature modeling capabilities between single-channel and multi-channel sequences.This series of experiments helps to clarify the applicability of different models across various datasets and variable scales, offering more targeted solutions for time series forecasting tasks.

To comprehensively evaluate the performance of the proposed model, it was also compared with the representative multiresolution method, N-HiTS, and the representative frequency-based methods, FEDformer and FiLM, on the ETTm2, Electricity, Traffic, and Weather datasets. The experimental results are shown in Table 9.

*Table 9.* Additional Comparative Experiments(MSE/MAE).

| Input Length | Timestacker | N-HiTS | FEDformer | FiLM |
|---|---|---|---|---|
| ETTm2 | 0.274/0.316 | 0.279/0.330 | 0.305/0.349 | 0.287/0.329 |
| Electricity | 0.194/0.275 | 0.186/0.287 | 0.214/0.327 | 0.223/0.302 |
| Traffic | 0.508/0.335 | 0.452/0.311 | 0.610/0.376 | 0.637/0.384 |
| Weather | 0.243/0.264 | 0.249/0.274 | 0.309/0.360 | 0.271/0.291 |

## E. SHOW CASE

A visualization of TimeStacker's prediction results was conducted across all datasets. As illustrated in Figure 6, in the 96-to-96 forecasting task, TimeStacker exhibited consistent performance across different datasets, clearly demonstrating its superior predictive capability.

To further highlight the capability of TimeStacker in capturing non-stationary signals, synthetic signals exhibiting nonlinear frequency variations over time were generated and evaluated alongside an MLP as a baseline. As illustrated in Figure 7, it can be observed that TimeStacker successfully captures high-frequency components, though slight phase misalignment is observed, whereas MLP-based models struggle to adapt to rapidly changing frequencies.

*Table 10.* Additional Experiments. All results are based on input sequences of length 96 and are calculated as the average across four different prediction lengths {96, 192, 336, 720}.

| Models | | **TimeStacker** | | SOFTS | | SparseTSF | | iTransformer | | TimeMixer | |
|---|---|---|---|---|---|---|---|---|---|---|---|
| Metric | | MSE | MAE | MSE | MAE | MSE | MAE | MSE | MAE | MSE | MAE |
| Traffic | 96 | 0.167 | 0.235 | 0.165 | 0.227 | 0.281 | 0.326 | 0.170 | 0.232 | 0.166 | 0.223 |
| | 192 | 0.157 | 0.223 | 0.158 | 0.223 | 0.236 | 0.282 | 0.158 | 0.22 | 0.157 | 0.230 |
| | 336 | 0.151 | 0.223 | 0.158 | 0.226 | 0.223 | 0.272 | 0.155 | 0.216 | 0.154 | 0.230 |
| | 720 | 0.169 | 0.242 | 0.174 | 0.244 | 0.242 | 0.292 | 0.182 | 0.254 | 0.171 | 0.242 |
| AVG | | **0.161** | 0.231 | 0.164 | **0.230** | 0.246 | 0.293 | 0.166 | 0.231 | 0.162 | 0.231 |
| Electricity | 96 | 0.296 | 0.384 | 0.292 | 0.381 | 0.477 | 0.504 | 0.299 | 0.4 | 0.297 | 0.391 |
| | 192 | 0.304 | 0.392 | 0.301 | 0.389 | 0.457 | 0.493 | 0.299 | 0.391 | 0.314 | 0.401 |
| | 336 | 0.364 | 0.42 | 0.365 | 0.418 | 0.489 | 0.512 | 0.362 | 0.426 | 0.367 | 0.431 |
| | 720 | 0.425 | 0.473 | 0.427 | 0.473 | 0.501 | 0.517 | 0.426 | 0.477 | 0.424 | 0.46 |
| AVG | | 0.347 | 0.417 | **0.346** | **0.415** | 0.481 | 0.507 | 0.347 | 0.424 | 0.351 | 0.421 |

*Table 11.* Impact of Look-Back Lengths. The backtracking window lengths were set to {96, 192, 336, 720}, and the prediction lengths were set to {96, 192, 336, 720}. The evaluation metrics used were MSE and MAE. The best performance of the model is highlighted in red.

| DataSet | | Exchange | | | | | | | | ETTh1 | | | | | | | | ETTh2 | | | | | | | |
|---|---|---|---|---|---|---|---|---|---|---|---|---|---|---|---|---|---|---|---|---|---|---|---|---|---|
| Look-Back Lengths | | 96 | | 192 | | 336 | | 720 | | 96 | | 192 | | 336 | | 720 | | 96 | | 192 | | 336 | | 720 | |
| Metric | | MSE | MAE | MSE | MAE | MSE | MAE | MSE | MAE | MSE | MAE | MSE | MAE | MSE | MAE | MSE | MAE | MSE | MAE | MSE | MAE | MSE | MAE | MSE | MAE |
| TimeMixer | 96 | 0.082 | 0.199 | 0.090 | 0.210 | 0.089 | 0.211 | 0.120 | 0.251 | 0.375 | 0.400 | 0.378 | 0.389 | 0.370 | 0.393 | 0.366 | 0.391 | 0.289 | 0.341 | 0.286 | 0.338 | 0.276 | 0.334 | 0.339 | 0.387 |
| | 192 | 0.177 | 0.297 | 0.181 | 0.303 | 0.178 | 0.301 | 0.173 | 0.299 | 0.429 | 0.421 | 0.428 | 0.420 | 0.414 | 0.415 | 0.411 | 0.424 | 0.372 | 0.392 | 0.367 | 0.388 | 0.347 | 0.382 | 0.387 | 0.410 |
| | 336 | 0.324 | 0.408 | 0.329 | 0.418 | 0.323 | 0.416 | 0.335 | 0.420 | 0.484 | 0.458 | 0.463 | 0.464 | 0.432 | 0.427 | 0.434 | 0.440 | 0.386 | 0.414 | 0.394 | 0.413 | 0.365 | 0.401 | 0.374 | 0.408 |
| | 720 | 0.837 | 0.691 | 0.817 | 0.679 | 0.887 | 0.699 | 0.904 | 0.741 | 0.498 | 0.482 | 0.465 | 0.462 | 0.447 | 0.455 | 0.450 | 0.472 | 0.412 | 0.434 | 0.413 | 0.436 | 0.405 | 0.434 | 0.410 | 0.443 |
| AVG | | 0.355 | 0.399 | 0.354 | 0.403 | 0.369 | 0.407 | 0.383 | 0.428 | 0.447 | 0.440 | 0.434 | 0.434 | 0.416 | 0.423 | 0.415 | 0.432 | 0.365 | 0.395 | 0.365 | 0.394 | 0.348 | 0.388 | 0.378 | 0.412 |
| SOFTS | 96 | 0.084 | 0.201 | 0.088 | 0.209 | 0.088 | 0.211 | 0.102 | 0.233 | 0.381 | 0.399 | 0.385 | 0.405 | 0.390 | 0.406 | 0.393 | 0.417 | 0.297 | 0.347 | 0.302 | 0.349 | 0.293 | 0.354 | 0.312 | 0.313 |
| | 192 | 0.172 | 0.294 | 0.181 | 0.303 | 0.190 | 0.315 | 0.183 | 0.309 | 0.435 | 0.431 | 0.431 | 0.432 | 0.428 | 0.432 | 0.429 | 0.438 | 0.373 | 0.394 | 0.387 | 0.407 | 0.383 | 0.407 | 0.386 | 0.415 |
| | 336 | 0.324 | 0.412 | 0.314 | 0.413 | 0.339 | 0.430 | 0.377 | 0.454 | 0.480 | 0.452 | 0.451 | 0.442 | 0.439 | 0.443 | 0.457 | 0.458 | 0.410 | 0.426 | 0.399 | 0.419 | 0.385 | 0.416 | 0.404 | 0.424 |
| | 720 | 0.811 | 0.672 | 0.732 | 0.645 | 0.815 | 0.690 | 0.845 | 0.701 | 0.499 | 0.488 | 0.444 | 0.460 | 0.450 | 0.468 | 0.463 | 0.477 | 0.411 | 0.433 | 0.427 | 0.444 | 0.435 | 0.454 | 0.429 | 0.455 |
| AVG | | 0.348 | 0.395 | 0.329 | 0.393 | 0.358 | 0.413 | 0.377 | 0.424 | 0.449 | 0.443 | 0.428 | 0.435 | 0.427 | 0.437 | 0.436 | 0.448 | 0.373 | 0.400 | 0.379 | 0.405 | 0.374 | 0.408 | 0.383 | 0.402 |
| iTransformer | 96 | 0.086 | 0.206 | 0.090 | 0.211 | 0.096 | 0.220 | 0.109 | 0.237 | 0.386 | 0.405 | 0.387 | 0.402 | 0.394 | 0.407 | 0.390 | 0.413 | 0.297 | 0.349 | 0.298 | 0.347 | 0.296 | 0.352 | 0.309 | 0.370 |
| | 192 | 0.177 | 0.299 | 0.193 | 0.316 | 0.192 | 0.319 | 0.193 | 0.319 | 0.441 | 0.436 | 0.446 | 0.429 | 0.444 | 0.433 | 0.418 | 0.431 | 0.380 | 0.400 | 0.383 | 0.399 | 0.389 | 0.405 | 0.390 | 0.415 |
| | 336 | 0.331 | 0.417 | 0.345 | 0.433 | 0.373 | 0.453 | 0.377 | 0.456 | 0.487 | 0.458 | 0.459 | 0.439 | 0.446 | 0.445 | 0.441 | 0.449 | 0.428 | 0.432 | 0.396 | 0.418 | 0.396 | 0.416 | 0.403 | 0.431 |
| | 720 | 0.847 | 0.691 | 0.757 | 0.657 | 0.694 | 0.645 | 0.796 | 0.683 | 0.503 | 0.491 | 0.464 | 0.468 | 0.467 | 0.463 | 0.489 | 0.490 | 0.427 | 0.445 | 0.422 | 0.441 | 0.426 | 0.447 | 0.450 | 0.465 |
| AVG | | 0.360 | 0.403 | 0.346 | 0.404 | 0.339 | 0.409 | 0.369 | 0.424 | 0.454 | 0.448 | 0.439 | 0.435 | 0.438 | 0.437 | 0.435 | 0.446 | 0.383 | 0.407 | 0.375 | 0.401 | 0.377 | 0.405 | 0.388 | 0.420 |
| SAMformer | 96 | 0.088 | 0.209 | 0.093 | 0.212 | 0.098 | 0.221 | 0.122 | 0.254 | 0.383 | 0.392 | 0.388 | 0.402 | 0.385 | 0.403 | 0.389 | 0.412 | 0.289 | 0.338 | 0.347 | 0.293 | 0.283 | 0.345 | 0.279 | 0.346 |
| | 192 | 0.176 | 0.299 | 0.186 | 0.307 | 0.196 | 0.318 | 0.229 | 0.339 | 0.438 | 0.423 | 0.438 | 0.429 | 0.419 | 0.424 | 0.425 | 0.436 | 0.401 | 0.398 | 0.382 | 0.401 | 0.36 | 0.397 | 0.36 | 0.396 |
| | 336 | 0.322 | 0.414 | 0.333 | 0.42 | 0.347 | 0.431 | 0.385 | 0.452 | 0.475 | 0.445 | 0.461 | 0.442 | 0.451 | 0.446 | 0.456 | 0.455 | 0.419 | 0.428 | 0.395 | 0.421 | 0.364 | 0.406 | 0.365 | 0.409 |
| | 720 | 0.799 | 0.673 | 0.804 | 0.678 | 0.856 | 0.701 | 0.957 | 0.741 | 0.478 | 0.468 | 0.464 | 0.469 | 0.454 | 0.468 | 0.467 | 0.482 | 0.421 | 0.44 | 0.415 | 0.441 | 0.405 | 0.438 | 0.4 | 0.439 |
| AVG | | 0.346 | 0.399 | 0.354 | 0.404 | 0.374 | 0.418 | 0.423 | 0.447 | 0.444 | 0.432 | 0.438 | 0.436 | 0.427 | 0.435 | 0.434 | 0.446 | 0.383 | 0.401 | 0.385 | 0.389 | 0.353 | 0.397 | 0.351 | 0.398 |
| **TimeStacker** | 96 | 0.084 | 0.200 | 0.084 | 0.201 | 0.085 | 0.204 | 0.085 | 0.205 | 0.379 | 0.385 | 0.37 | 0.384 | 0.362 | 0.383 | 0.364 | 0.383 | 0.28 | 0.327 | 0.283 | 0.332 | 0.27 | 0.333 | 0.267 | 0.331 |
| | 192 | 0.171 | 0.293 | 0.173 | 0.293 | 0.174 | 0.294 | 0.171 | 0.293 | 0.429 | 0.416 | 0.424 | 0.414 | 0.401 | 0.409 | 0.399 | 0.418 | 0.373 | 0.385 | 0.366 | 0.386 | 0.348 | 0.383 | 0.336 | 0.374 |
| | 336 | 0.314 | 0.408 | 0.317 | 0.411 | 0.32 | 0.413 | 0.321 | 0.408 | 0.459 | 0.436 | 0.448 | 0.429 | 0.428 | 0.427 | 0.42 | 0.428 | 0.407 | 0.416 | 0.397 | 0.413 | 0.362 | 0.397 | 0.354 | 0.396 |
| | 720 | 0.776 | 0.656 | 0.756 | 0.644 | 0.792 | 0.658 | 0.772 | 0.656 | 0.464 | 0.455 | 0.447 | 0.453 | 0.432 | 0.45 | 0.427 | 0.447 | 0.412 | 0.431 | 0.409 | 0.428 | 0.399 | 0.428 | 0.389 | 0.426 |
| AVG | | **0.336** | **0.389** | 0.333 | **0.387** | 0.343 | **0.392** | **0.337** | **0.391** | **0.433** | **0.423** | **0.422** | **0.420** | **0.406** | **0.417** | **0.403** | **0.419** | 0.368 | **0.390** | **0.364** | 0.390 | **0.345** | **0.385** | **0.337** | **0.382** |

*Table 12.* Complete Experimental Result. All results are based on input sequences of length 96 and are calculated as the average across four different prediction lengths {96, 192, 336, 720}.

| Models | | TimeStacker (ours) | | SOFTS (2024) | | SparseTSF (2024) | | iTransformer (2024) | | TimeMixer (2024) | | SAMformer (2024) | | PatchTST (2023) | | Crossformer (2023) | | DLinear (2023) | | RLinear (2023) | |
|---|---|---|---|---|---|---|---|---|---|---|---|---|---|---|---|---|---|---|---|---|---|
| **Metric** | | MSE | MAE | MSE | MAE | MSE | MAE | MSE | MAE | MSE | MAE | MSE | MAE | MSE | MAE | MSE | MAE | MSE | MAE | MSE | MAE |
| ETTh1 | 96 | 0.379 | **0.385** | 0.381 | 0.399 | 0.388 | 0.387 | 0.386 | 0.405 | **0.375** | 0.400 | 0.383 | 0.392 | 0.414 | 0.419 | 0.423 | 0.448 | 0.386 | 0.400 | 0.386 | 0.395 |
| | 192 | **0.429** | **0.416** | 0.435 | 0.431 | 0.438 | 0.417 | 0.441 | 0.436 | **0.429** | 0.421 | 0.438 | 0.423 | 0.460 | 0.445 | 0.471 | 0.474 | 0.437 | 0.432 | 0.437 | 0.424 |
| | 336 | **0.459** | **0.436** | 0.480 | 0.452 | 0.469 | 0.438 | 0.487 | 0.458 | 0.484 | 0.458 | 0.475 | 0.445 | 0.501 | 0.466 | 0.570 | 0.546 | 0.481 | 0.459 | 0.479 | 0.446 |
| | 720 | **0.464** | **0.455** | 0.499 | 0.488 | 0.468 | 0.457 | 0.503 | 0.491 | 0.498 | 0.482 | 0.478 | 0.468 | 0.500 | 0.488 | 0.653 | 0.621 | 0.519 | 0.516 | 0.481 | 0.470 |
| AVG | | **0.433** | **0.423** | 0.449 | 0.442 | 0.441 | 0.425 | 0.454 | 0.447 | 0.447 | 0.440 | 0.444 | 0.432 | 0.469 | 0.454 | 0.529 | 0.522 | 0.456 | 0.452 | 0.446 | 0.434 |
| ETTh2 | 96 | **0.280** | **0.327** | 0.297 | 0.347 | 0.304 | 0.346 | 0.297 | 0.349 | 0.289 | 0.341 | 0.289 | 0.338 | 0.302 | 0.348 | 0.745 | 0.584 | 0.333 | 0.387 | 0.288 | 0.338 |
| | 192 | 0.373 | 0.385 | 0.373 | 0.394 | 0.409 | 0.403 | 0.380 | 0.400 | **0.372** | **0.392** | 0.401 | 0.398 | 0.388 | 0.400 | 0.877 | 0.656 | 0.477 | 0.476 | 0.374 | 0.390 |
| | 336 | 0.407 | 0.416 | 0.410 | 0.426 | 0.426 | 0.430 | 0.428 | 0.432 | **0.386** | **0.414** | 0.419 | 0.428 | 0.426 | 0.433 | 1.043 | 0.731 | 0.594 | 0.541 | 0.415 | 0.426 |
| | 720 | 0.412 | **0.431** | 0.411 | 0.433 | 0.421 | 0.438 | 0.427 | 0.445 | 0.412 | 0.434 | 0.421 | 0.440 | 0.431 | 0.446 | 1.104 | 0.763 | 0.831 | 0.657 | 0.420 | 0.440 |
| AVG | | 0.368 | **0.390** | 0.373 | 0.400 | 0.421 | 0.438 | 0.383 | 0.407 | **0.364** | 0.395 | 0.383 | 0.401 | 0.387 | 0.407 | 0.942 | 0.684 | 0.559 | 0.515 | 0.374 | 0.398 |
| ETTm1 | 96 | **0.311** | **0.337** | 0.325 | 0.361 | 0.366 | 0.369 | 0.334 | 0.368 | 0.320 | 0.357 | 0.352 | 0.374 | 0.329 | 0.367 | 0.404 | 0.426 | 0.345 | 0.372 | 0.355 | 0.376 |
| | 192 | 0.364 | **0.367** | 0.375 | 0.389 | 0.404 | 0.387 | 0.377 | 0.391 | **0.361** | 0.381 | 0.392 | 0.392 | 0.367 | 0.385 | 0.450 | 0.451 | 0.380 | 0.389 | 0.391 | 0.392 |
| | 336 | **0.389** | **0.391** | 0.405 | 0.412 | 0.432 | 0.406 | 0.426 | 0.420 | 0.390 | 0.404 | 0.425 | 0.413 | 0.399 | 0.410 | 0.532 | 0.515 | 0.413 | 0.413 | 0.424 | 0.415 |
| | 720 | 0.460 | **0.428** | 0.466 | 0.447 | 0.496 | 0.442 | 0.491 | 0.459 | **0.454** | 0.441 | 0.49 | 0.449 | 0.454 | 0.439 | 0.666 | 0.589 | 0.474 | 0.453 | 0.487 | 0.450 |
| AVG | | **0.381** | **0.381** | 0.393 | 0.403 | 0.425 | 0.401 | 0.407 | 0.410 | **0.381** | 0.395 | 0.415 | 0.407 | 0.387 | 0.400 | 0.513 | 0.496 | 0.403 | 0.407 | 0.414 | 0.407 |
| ETTm2 | 96 | **0.171** | **0.250** | 0.180 | 0.261 | 0.198 | 0.272 | 0.180 | 0.264 | 0.175 | 0.258 | 0.181 | 0.264 | 0.175 | 0.259 | 0.287 | 0.366 | 0.193 | 0.292 | 0.182 | 0.265 |
| | 192 | **0.235** | **0.292** | 0.246 | 0.306 | 0.259 | 0.308 | 0.250 | 0.309 | 0.237 | 0.299 | 0.245 | 0.305 | 0.241 | 0.302 | 0.414 | 0.492 | 0.284 | 0.362 | 0.246 | 0.304 |
| | 336 | **0.293** | **0.329** | 0.319 | 0.352 | 0.315 | 0.343 | 0.311 | 0.348 | 0.298 | 0.340 | 0.305 | 0.341 | 0.305 | 0.343 | 0.597 | 0.542 | 0.369 | 0.427 | 0.307 | 0.342 |
| | 720 | 0.395 | **0.391** | 0.405 | 0.401 | 0.416 | 0.399 | 0.412 | 0.407 | **0.391** | 0.396 | 0.409 | 0.398 | 0.402 | 0400 | 1.730 | 1.042 | 0.554 | 0.522 | 0.407 | 0.398 |
| AVG | | **0.274** | **0.316** | 0.287 | 0.330 | 0.297 | 0.331 | 0.288 | 0.332 | 0.275 | 0.323 | 0.285 | 0.327 | 0.281 | 0.326 | 0.757 | 0.610 | 0.350 | 0.401 | 0.286 | 0.327 |
| Traffic | 96 | 0.496 | 0.331 | **0.376** | **0.251** | 0.559 | 0.335 | 0.395 | 0.268 | 0.462 | 0.285 | 0.552 | 0.367 | 0.462 | 0.295 | 0.522 | 0.290 | 0.650 | 0.396 | 0.649 | 0.389 |
| | 192 | 0.491 | 0.331 | **0.398** | **0.261** | 0.567 | 0.346 | 0.417 | 0.276 | 0.473 | 0.296 | 0.569 | 0.368 | 0.466 | 0.296 | 0.530 | 0.293 | 0.598 | 0.370 | 0.601 | 0.366 |
| | 336 | 0.505 | 0.334 | **0.415** | **0.269** | 0.575 | 0.349 | 0.433 | 0.283 | 0.498 | 0.296 | 0.586 | 0.376 | 0.482 | 0.304 | 0.558 | 0.305 | 0.605 | 0.373 | 0.609 | 0.369 |
| | 720 | 0.541 | 0.343 | **0.447** | **0.287** | 0.609 | 0.368 | 0.467 | 0.302 | 0.506 | 0.313 | 0.63 | 0.401 | 0.514 | 0.322 | 0.589 | 0.328 | 0.645 | 0.394 | 0.647 | 0.387 |
| AVG | | 0.508 | 0.335 | **0.409** | **0.267** | 0.578 | 0.350 | 0.428 | 0.282 | 0.484 | 0.297 | 0.595 | 0.382 | 0.481 | 0.304 | 0.550 | 0.304 | 0.625 | 0.383 | 0.626 | 0.378 |
| Electricity | 96 | 0.168 | 0.251 | **0.143** | **0.233** | 0.202 | 0.261 | 0.148 | 0.240 | 0.153 | 0.247 | 0.199 | 0.277 | 0.181 | 0.270 | 0.219 | 0.314 | 0.197 | 0.282 | 0.201 | 0.281 |
| | 192 | 0.176 | 0.262 | **0.158** | **0.248** | 0.207 | 0.277 | 0.162 | 0.253 | 0.166 | 0.256 | 0.199 | 0.279 | 0.188 | 0.284 | 0.231 | 0.322 | 0.196 | 0.285 | 0.201 | 0.283 |
| | 336 | 0.195 | 0.278 | **0.178** | **0.269** | 0.219 | 0.292 | 0.178 | 0.269 | 0.185 | 0.277 | 0.214 | 0.294 | 0.204 | 0.293 | 0.246 | 0.337 | 0.209 | 0.301 | 0.215 | 0.298 |
| | 720 | 0.235 | 0.310 | **0.218** | **0.305** | 0.261 | 0.324 | 0.225 | 0.317 | 0.225 | 0.310 | 0.257 | 0.328 | 0.246 | 0.324 | 0.280 | 0.363 | 0.245 | 0.333 | 0.257 | 0.331 |
| AVG | | 0.194 | 0.275 | **0.174** | **0.264** | 0.222 | 0.289 | 0.178 | 0.270 | 0.182 | 0.272 | 0.217 | 0.295 | 0.205 | 0.290 | 0.244 | 0.334 | 0.212 | 0.300 | 0.219 | 0.298 |
| Weather | 96 | 0.161 | **0.198** | 0.166 | 0.208 | 0.213 | 0.250 | 0.174 | 0.214 | 0.163 | 0.209 | 0.193 | 0.205 | 0.177 | 0.218 | **0.158** | 0.230 | 0.196 | 0.255 | 0.192 | 0.232 |
| | 192 | 0.207 | **0.241** | 0.217 | 0.253 | 0.259 | 0.282 | 0.221 | 0.254 | 0.208 | 0.250 | 0.242 | 0.274 | 0.225 | 0.259 | **0.206** | 0.277 | 0.237 | 0.296 | 0.240 | 0.271 |
| | 336 | 0.261 | **0.281** | 0.282 | 0.300 | 0.308 | 0.315 | 0.278 | 0.296 | 0.251 | 0.287 | 0.284 | 0.309 | 0.278 | 0.297 | 0.272 | 0.335 | 0.283 | 0.335 | 0.292 | 0.307 |
| | 720 | 0.343 | **0.334** | 0.356 | 0.351 | 0.380 | 0.361 | 0.358 | 0.347 | **0.339** | 0.341 | 0.358 | 0.351 | 0.354 | 0.348 | 0.398 | 0.418 | 0.345 | 0.381 | 0.364 | 0.353 |
| AVG | | 0.243 | **0.264** | 0.255 | 0.278 | 0.290 | 0.302 | 0.258 | 0.278 | **0.240** | 0.271 | 0.264 | 0.285 | 0.259 | 0.348 | 0.259 | 0.315 | 0.265 | 0.317 | 0.272 | 0.291 |
| Exchange | 96 | 0.084 | 0.200 | 0.084 | 0.201 | 0.093 | 0.217 | 0.086 | 0.206 | **0.082** | **0.199** | 0.088 | 0.209 | 0.088 | 0.205 | 0.256 | 0.367 | 0.088 | 0.218 | 0.093 | 0.217 |
| | 192 | **0.171** | **0.293** | 0.172 | 0.294 | 0.179 | 0.304 | 0.177 | 0.299 | 0.177 | 0.297 | 0.176 | 0.299 | 0.176 | 0.299 | 0.470 | 0.509 | 0.176 | 0.315 | 0.184 | 0.307 |
| | 336 | 0.314 | 0.406 | 0.324 | 0.412 | 0.319 | 0.410 | 0.331 | 0.417 | 0.324 | 0.408 | 0.322 | 0.414 | **0.301** | **0.397** | 1.268 | 0.883 | 0.313 | 0.427 | 0.351 | 0.432 |
| | 720 | **0.776** | **0.656** | 0.811 | 0.672 | 0.823 | 0.683 | 0.847 | 0.691 | 0.837 | 0.691 | 0.799 | 0.673 | 0.901 | 0.714 | 1.767 | 1.068 | 0.839 | 0.695 | 0.886 | 0.714 |
| AVG | | **0.336** | **0.389** | 0.348 | 0.395 | 0.365 | 0.401 | 0.360 | 0.403 | 0.355 | 0.399 | 0.346 | 0.399 | 0.367 | 0.404 | 0.940 | 0.707 | 0.354 | 0.414 | 0.378 | 0.417 |
| 1st Count | | **15** | **26** | 11 | 10 | 0 | 0 | 0 | 0 | 13 | 4 | 0 | 0 | 2 | 1 | 2 | 0 | 0 | 0 | 0 | 0 |

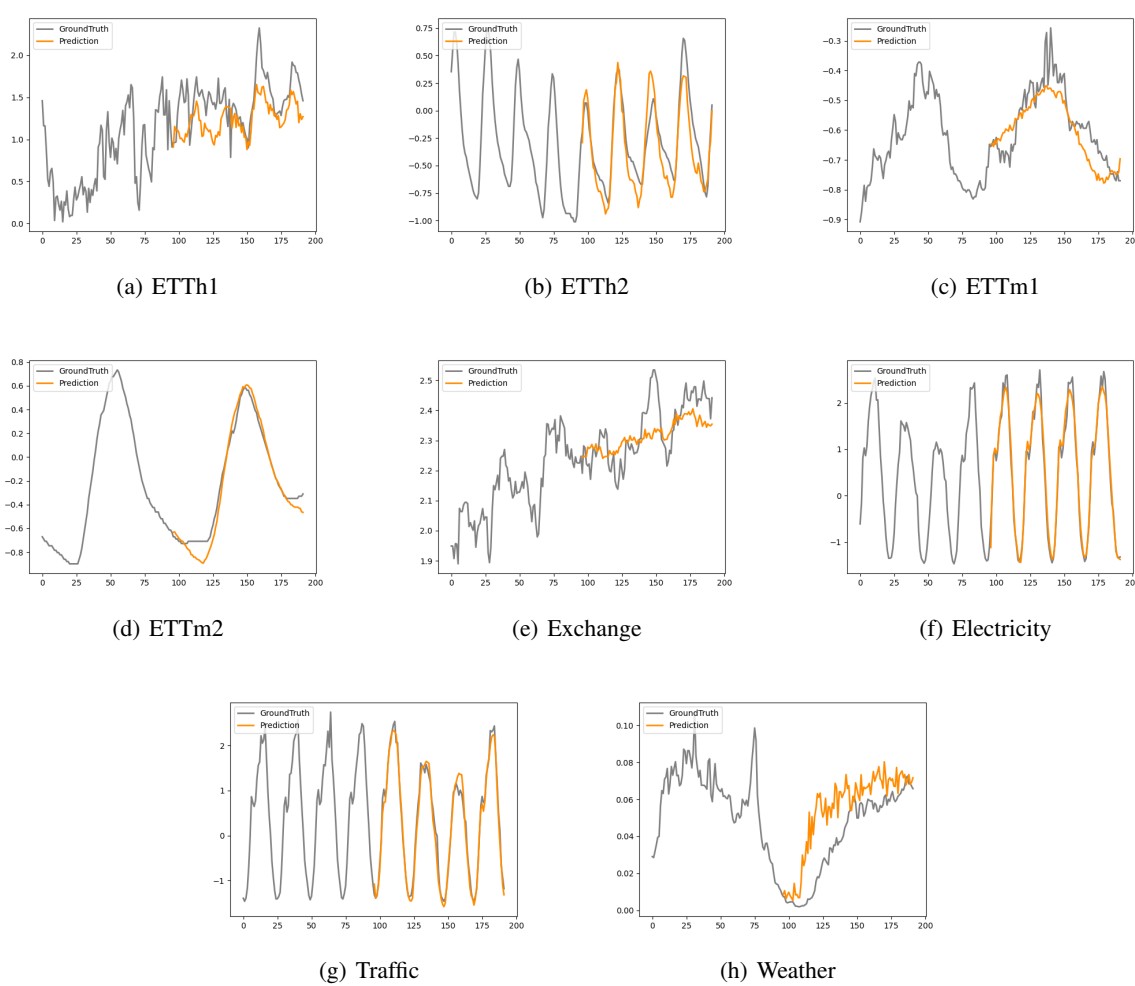

*Figure 6.* Visualization of forecasting results on the real-world dataset with look-back window length $L = 96$ and prediction length $H = 96$

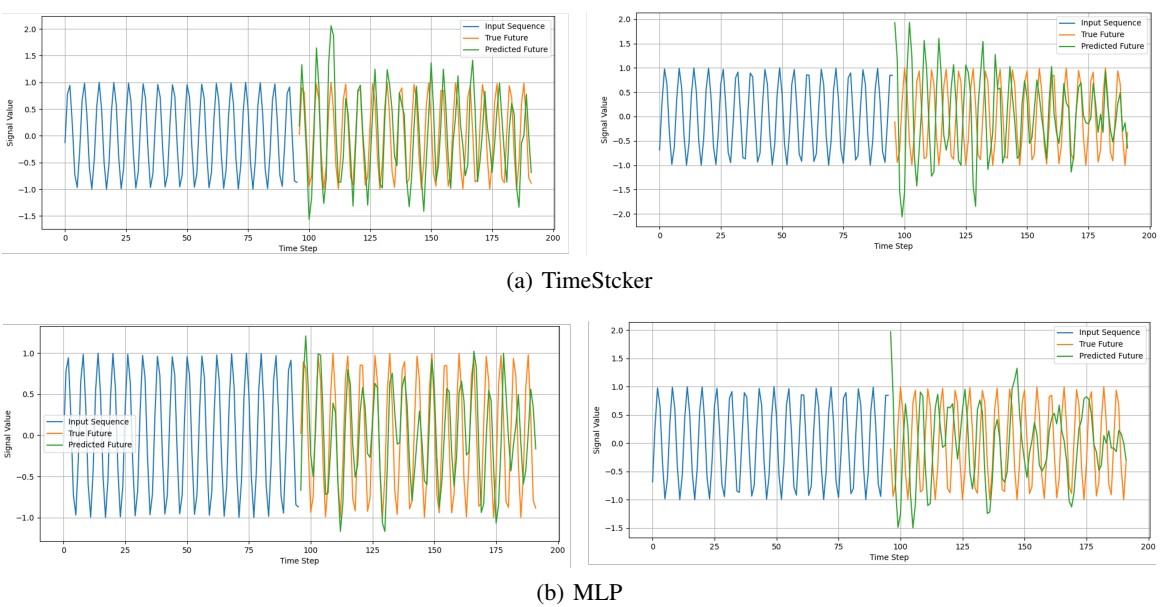

(a) TimeStcker

(b) MLP

*Figure 7.* Visualization of results on synthetic data. Non-stationary signals with time-varying frequencies were used for training and testing. (a) presents the visualization produced by TimeStacker, while (b) presents the visualization produced by the MLP.

