# OpenReview forum: "TimeStacker: A Novel Framework with Multilevel Observation for Capturing Nonstationary Patterns in Time Series Forecasting"
_ICML.cc/2025/Conference — ICML 2025 poster_

### Official Review · Reviewer_PxiA · 2025-03-07

**Overall Recommendation:** 3

**Summary:**

The paper introduces TimeStacker, a new framework designed to enhance time series forecasting by effectively capturing nonstationary patterns. The core innovation lies in its stacking mechanism, which sequentially aggregates patches of varying sizes to balance global and local signal representations. Additionally, the framework employs a frequency-based self-attention module that improves feature modeling by computing similarity in the frequency domain while aggregating in the time domain. Experimental results across multiple real-world datasets (energy, finance, weather) demonstrate that TimeStacker achieves state-of-the-art performance, surpassing existing models in both predictive accuracy and computational efficiency while using fewer parameters.

**Claims And Evidence:**

The claims made in the submission are largely supported, through experiments on multiple datasets. The results consistently demonstrate that TimeStacker outperforms state-of-the-art models in predictive accuracy while maintaining computational efficiency. The inclusion of ablation studies further strengthens the validity of the proposed frequency-based self-attention mechanism. But, one potential limitation is the claim that TimeStacker effectively handles multivariate time series. While the model performs well on most datasets, the authors acknowledge a decline in performance as the number of variables increases, suggesting a possible bottleneck. Additionally, while the theoretical justification of the stacking mechanism is well-founded, more in-depth comparisons with alternative approaches for handling nonstationary signals could strengthen the argument. Overall, the empirical results are compelling, but further validation on larger and more complex datasets would reinforce the generalizability of the proposed method.

**Essential References Not Discussed:**

No major reference missing

**Experimental Designs Or Analyses:**

The evaluation framework is well-structured, with comparisons against multiple state-of-the-art models across diverse benchmark datasets, ensuring assessment of TimeStacker’s performance. The use of standard forecasting metrics, MSE and MAE, that supports the reliability of the results. Additionally, the ablation studies provide insights into the contribution of the frequency-based self-attention module. But, a deeper analysis of computational complexity trade-offs compared to baseline models would strengthen the argument. Overall, the experimental design is good, but more investigation into scalability and complexity would enhance the paper.

**Methods And Evaluation Criteria:**

The methods and evaluation criteria align well with the problem of time series forecasting. TimeStacker introduces a novel stacking mechanism and frequency-based self-attention, both of which are well-motivated by the challenges of nonstationary signals. The choice of benchmark datasets ensures a comprehensive evaluation across diverse real-world applications. The use of Mean Squared Error (MSE) and Mean Absolute Error (MAE) as performance metrics is standard in time series forecasting and appropriately assesses both the accuracy and robustness of predictions. Ablation studies and comparisons with recent state-of-the-art models provide further validation. While the evaluation framework is well-structured, the model's performance on highly multivariate datasets could have been explored further to assess scalability.

**Other Comments Or Suggestions:**

Table 1, could benefit from having a average across datasets to see overall improvement compared.

**Other Strengths And Weaknesses:**

No additional points

**Questions For Authors:**

No additional questions

**Relation To Broader Scientific Literature:**

The proposed TimeStacker framework is related to prior work in deep learning-based forecasting models, such as MLP-based approaches (DLinear, TimeMixer) and Transformer-based models (PatchTST, Crossformer). The contribution connects with broader trends in time-frequency analysis and multi-scale modeling, which have been explored in statistical and signal processing literature. By integrating these concepts into a computationally efficient deep learning framework, the paper advances the field by offering a scalable and interpretable solution for nonstationary time series forecasting.

**Theoretical Claims:**

The authors reference the time-frequency uncertainty principle to justify their multi-scale stacking approach, which is a well-established concept in signal processing. The mathematical formulation of the FreqAttention module, including the use of Fourier transforms and Hadamard products for computing similarity, appears logically sound and aligns with existing principles in time-series analysis.

---

> ### Author Rebuttal · Authors · 2025-03-31
>
> We thank the reviewer for your appreciation and  for the valuable suggestions.
>
> **Cross-Dataset Performance:**
>
> For the ETT series data, we present the average performance across datasets (the first row indicates MSE and the second row indicates MAE) as shown below:
>
> | TimeStaker | SOFTS | SparseTSF | iTransformer | TimeMixer | SAMformer | PatchTST | Crossformer | DLinear | RLinear |
> | ---------- | ----- | --------- | ------------ | --------- | --------- | -------- | ----------- | ------- | ------- |
> | 0.364      | 0.376 | 0.396     | 0.383        | 0.367     | 0.382     | 0.381    | 0.685       | 0.442   | 0.380   |
> | 0.378      | 0.394 | 0.398     | 0.399        | 0.388     | 0.392     | 0.397    | 0.578       | 0.444   | 0.392   |
>
>
>
> **Complexity Analysis:**
>
> We have conducted an in-depth analysis on how increasing the input sequence length affects memory consumption and training time, with the results averaged over three runs. Partial results are shown in the table below (GPU Memory(MB)/Training Time(ms/iter):
>
> | Input Length | TimeStaker | SparseTSF | PatchTST   | Crossformer |
> | ------------ | ---------- | --------- | ---------- | ----------- |
> | 192          | 28.8/134   | 15.6/108  | 145.8/87.8 | 5214/238    |
> | 384          | 29.3/133   | 16.1/112  | 334.7/90.1 | 5734/273    |
> | 768          | 34.3/137   | 20.6/115  | 830.0/93.3 | 6814/342    |
> | 1536         | 59.2/137   | 28.7/127  | 2403.6/137 | 9016/1007   |
> | 3072         | 110.7/134  | 44.1/131  | 7832/1315  | 12470/3121  |
>
> We will provide clearer visualizations and more comprehensive data in the appendix.
>
>
>
> **Regarding Multivariate Data:**
>
> Our work primarily focuses on time series modeling and does not include specialized designs for multivariate data. Integrating information from multiple variables is a complex task that we plan to address in our future work.
>
> Once again, thank you for your valuable feedback.

---

> > ### Comment · Reviewer_PxiA · 2025-04-08
> >
> > Thank you for answering the questions and taking the feedback into consideration. I would like to keep my original score.

---

### Official Review · Reviewer_fBGw · 2025-03-12

**Overall Recommendation:** 2

**Summary:**

The paper "TimeStacker: A Novel Framework with Multilevel Observation for Capturing Nonstationary Patterns in Time Series Forecasting" introduces TimeStacker, a forecasting framework that addresses the challenges of nonstationary time series by integrating multi-resolution stacking and frequency-based self-attention. By sequentially aggregating patches of varying sizes, TimeStacker captures both global trends and local variations, while its frequency-based attention module enhances feature extraction by computing similarity in the frequency domain. Grounded in time-frequency analysis and the uncertainty principle, the model outperforms state-of-the-art forecasting methods across diverse real-world datasets, achieving superior accuracy with lower computational complexity.

## After rebuttal

 I will maintain my score

**Claims And Evidence:**

The paper asserts the necessity of analyzing time series in the frequency domain and adopting a multi-resolution perspective, claims that are well-supported by **Definition 3.1 and Theorem 3.2**, which theoretically establish the importance of capturing time-frequency variations in nonstationary signals. Furthermore, the experimental results empirically validate these claims by demonstrating TimeStacker’s superior performance across multiple real-world datasets. However, while the proposed approach effectively addresses the stated challenges, it bears resemblance to existing methodologies that employ similar multi-resolution strategies, which somewhat limits its novelty.

**Essential References Not Discussed:**

Challu, Cristian et al. “N-HiTS: Neural Hierarchical Interpolation for Time Series Forecasting.” AAAI 2023

**Experimental Designs Or Analyses:**

The experimental design presented in the paper is generally well-structured and valid, employing widely recognized benchmarks and evaluation metrics commonly used in time-series forecasting. The results effectively demonstrate the advantages of the proposed method. However, to further strengthen the empirical analysis, it would be beneficial to include an ablation study on the multi-resolution stacking steps, providing insights into how each resolution level contributes to the final performance. Additionally, a direct performance comparison with N-HiTS, which also utilizes a hierarchical decomposition strategy, would help clarify TimeStacker’s relative advantages and better position it within the landscape of multi-resolution forecasting models.

**Methods And Evaluation Criteria:**

The proposed method adopts a multi-resolution approach to analyzing time series in the frequency domain, which is a well-motivated strategy for handling nonstationary signals. However, to strengthen the contribution, a clearer distinction between TimeStacker and existing models such as N-HiTS and TimeMixer—which also leverage multi-resolution techniques—would be beneficial. Regarding the evaluation criteria, the paper employs widely accepted metrics (e.g., MSE, MAE) and benchmark datasets commonly used in time-series forecasting research, ensuring a fair and standardized comparison against existing methods.

**Other Comments Or Suggestions:**

Some figures appear blurry and lack clarity, making it difficult to interpret fine details. Improving the resolution and contrast of the figures would enhance readability.

**Other Strengths And Weaknesses:**

**Strengths:**
1. The paper is well-structured and intuitive, making it easy to follow. The proposed multi-resolution stacking and frequency-based self-attention mechanisms are clearly explained, allowing readers to grasp the motivation behind TimeStacker without excessive complexity.
2. The mathematical foundations provided through Definition 3.1 and Theorem 3.2 effectively support the proposed approach, strengthening its conceptual motivation and distinguishing it from purely empirical contributions.

 **Weaknesses:**
1. While the paper presents a well-motivated framework, its core ideas (multi-resolution analysis and frequency-domain modeling) are not fundamentally new, as similar approaches have been explored in models like **N-HiTS** and Fourier-based architectures (e.g., FEDformer, FiLM). A clearer articulation of how TimeStacker differs from or improves upon these methods would enhance its originality.
2. The paper does not adequately differentiate itself from prior work that employs hierarchical decomposition and frequency modeling. Explicit comparisons—both in theoretical discussion and experimental evaluation—would strengthen the argument for its contribution.
3. Additional points regarding experimental design, missing citations, and potential improvements have been detailed in responses to individual review questions.

Overall, while the paper is well-written and theoretically grounded, addressing its novelty concerns and improving its positioning relative to prior multi-resolution models would make the contribution more compelling.

**Questions For Authors:**

1. How does TimeStacker differentiate itself from other multi-resolution approaches in time-series forecasting?
   - Several existing methods, such as N-HiTS and other hierarchical decomposition models, already leverage multi-resolution processing. Could you clearly articulate the key differences between TimeStacker and these models in terms of both methodology and performance?
   - A more explicit comparison could help clarify the novelty and contribution of the proposed approach.

2. In what scenarios does TimeStacker outperform existing frequency-domain-based models?
   - The paper emphasizes the advantages of analyzing time-series data in the frequency domain, but how does TimeStacker compare against prior frequency-aware models such as FEDformer, FiLM, or other Fourier-based architectures?
   - Are there specific types of datasets, forecasting horizons, or signal characteristics where TimeStacker’s design proves particularly effective? Including such insights would strengthen the empirical justification for the proposed method.

3. Does applying smoothness in the inter-patch frequency-based attention module conflict with the goal of capturing local features?
   - The smooth layer is stated to reduce noise within patches, yet patches themselves are meant to capture localized variations in the time series.
   - Would excessive smoothing risk removing important short-term patterns or distort fine-grained structures?
   - Could you provide any empirical justification or ablation studies showing how this trade-off impacts performance?

**Relation To Broader Scientific Literature:**

The paper introduces a novel perspective on time-series forecasting by emphasizing the importance of frequency-domain analysis and multi-resolution decomposition, challenging the traditional focus on purely temporal correlations. This approach aligns with prior research on multi-scale modeling (e.g., N-HiTS, TimeMixer) but distinguishes itself by explicitly leveraging frequency-based self-attention and progressive stacking to capture both global trends and local variations. By grounding its methodology in time-frequency analysis and the uncertainty principle, the paper contributes to a broader shift in the field, encouraging researchers to rethink conventional time-series modeling paradigms. While the proposed framework builds on existing ideas, it provides a cohesive and theoretically justified approach, which could inspire further advancements in handling nonstationary time-series data.

**Theoretical Claims:**

I have examined **Definition 3.1 and Theorem 3.2**, both of which serve as the theoretical foundation for TimeStacker’s motivation. **Definition 3.1** effectively formulates the need for frequency-domain analysis by representing nonstationary time series as time-varying Fourier components, while **Theorem 3.2**, derived from the time-frequency uncertainty principle, justifies the necessity of a multi-resolution approach. These theoretical claims are mathematically sound and align with well-established principles in signal processing. Additionally, they provide a clear rationale for the model’s design choices. I did not identify any fundamental issues with these proofs, but a deeper comparison with alternative formulations could further reinforce their validity.

---

> ### Author Rebuttal · Authors · 2025-03-31
>
> We thank the reviewer for your recognition of our work and your constructive feedback.
>
>
>
> **Q1&W1：**
>
> Our approach fundamentally differs from multi-resolution and Fourier‑based methods. While the latter emphasize extracting static features from various frequency bands—that is, observing the signal at different granularities—TimeStacker focuses on capturing the dynamic evolution within the input signal to reveal its underlying transformation patterns. For this reason, we refer to our method as **“multi‑level”** rather than **“multi‑resolution”**.
>
> In TimeStacker, a sequence
>
> $$
> X = \{x_1, x_2, x_3, ..., x_t\}
> $$
>
>  is first partitioned into K segments (ensuring that t is divisible by K), resulting in a new sequence
>
> $$
> \hat{X}_k= \{X_1, X_2, ..., X_{t/k}\},X_i = \{x_{(i-1)\times l + 1}, x_{(i-1)\times 1 + 2}, ..., x_{i\times l}\}, l=t/K
> $$
>
> Next, we compute the similarity between these subsequences in the **frequency domain**—essentially using a window of size K with stride K to observe the internal evolution of the sequence. Based on this similarity, we aggregate the subsequences in the **time domain** to produce an output sequence of length t. By continuously varying the window shape (i.e., reducing the observation window K) and repeating the process across multiple levels, we break the constraints imposed by Theorem 3.2 to capture the variation patterns of the Fourier coefficients *a* and *b* as defined in Definition 3.1.
>
>
>
> In contrast, multi-resolution methods such as N‑HiTS primarily extract signal features via downsampling. For example, given a sequence
>
> $$
> X = \{X_1, X_2, X_3, ..., X_t\},
> $$
>
>  downsampling with a stride of K produces a new sequence
>
>
> $$
> \hat{X}_k = \{x_1, x_{2k}, x_{3k}, ..., x_t\},
> $$
>
>
>  followed by interpolation for prediction. The process is then repeated with smaller strides to capture information at different granularities along the entire sequence.
>
>
>
> **Q2&W2&W3：**
>
> At a detailed level, TimeStacker leverages frequency‑domain information to observe the dynamic evolution of a sequence. Its process can be summarized as follows:
>
> > Sequence → Transformation (Frequency Domain) → Compute variation patterns (Fourier coefficients *a* and *b*) among subsequences → Aggregate in the Time Domain based on these patterns → Sequence → Prediction
>
> This approach also helps mitigate errors that can arise from the inverse transformation using discrete orthogonal bases.
>
> In contrast, other Fourier‑based architectures (e.g., FEDformer, FiLM) project the time‑domain sequence onto a Fourier (or other orthogonal) basis, enhance the features in the frequency domain, and then apply an inverse transformation to return to the time domain. Their process can be abstractly described as:
>
> > Sequence → Transformation (Frequency Domain) → Feature Enhancement → Inverse Transformation (Time Domain) → Sequence → Prediction
>
>
>
> Below is a preliminary comparison between TimeStacker and baseline models (N‑HiTS, FEDformer, and FiLM). The first value in each cell represents MSE and the second represents MAE. A more comprehensive comparison will be added to the main text and appendix.
>
> | Dataset     | TimeStacker   | N‑HiTS        | FEDformer     | FiLM          |
> | ----------- | ------------- | ------------- | ------------- | ------------- |
> | ETTm2       | 0.274 / 0.316 | 0.279 / 0.330 | 0.305 / 0.349 | 0.287 / 0.329 |
> | Electricity | 0.194 / 0.275 | 0.186 / 0.287 | 0.214 / 0.327 | 0.223 / 0.302 |
> | Traffic     | 0.508 / 0.335 | 0.452 / 0.311 | 0.610 / 0.376 | 0.637 / 0.384 |
> | Weather     | 0.243 / 0.264 | 0.249 / 0.274 | 0.309 / 0.360 | 0.271 / 0.291 |
>
>
>
> **Q3：**
>
> We define the output of our smoothing layer as ***SmoothLayer(x) + x***, as presented in Equation (13) of our paper, incorporating a residual connection. This mechanism ensures that even if *SmoothLayer(·)* excessively smooths the signal—thereby potentially losing local features—the residual branch can effectively compensate by reintroducing these features. Consequently, the approach minimizes noise interference while preserving key information, thus enhancing the model’s expressive power.
>
>
>
> **W3：**
>
> We appreciate your suggestions. We will add the relevant essential references to the main text and plan to release the code publicly in the near future. Additionally, we will include a complete comparison with N‑HiTS, FEDformer, and FiLM, and, based on the feedback from all reviewers, we will augment our experimental data to more comprehensively demonstrate the advantages of our approach and further enrich the paper.

---

> > ### Comment · Reviewer_fBGw · 2025-04-08
> >
> > I appreciate the authors' explanation and additional results. I believe that the additional empirical results provided in response to my questions and those of other reviewers should be included in the revised paper and would significantly strengthen it.  I respect the other reviewers' comments and I will maintain my score.

---

### Official Review · Reviewer_FZmL · 2025-03-13

**Overall Recommendation:** 2

**Summary:**

This paper is another incremental work in developing Transformer-based time-series architectures and follow some widely used yet problematic benchmarks, such as ETT, Exchange, Weather, etc.

**Claims And Evidence:**

This paper claims its proposed approach may better tackle non-stationary signals in time-series forecasting.

**Essential References Not Discussed:**

If this paper cares about non-stationarity, maybe some existing normalization methods should be mentioned, discussed, and compared if possible.

For example, can your proposed architectures handle non-stationarity without using RevIN [1]? If not, why the proposed modules tackle the non-stationarity challenge and how to demonstrate that?

[1] Kim, T., Kim, J., Tae, Y., Park, C., Choi, J.-H., and Choo, J. Reversible instance normalization for accurate time-series forecasting against distribution shift. In International Conference on Learning Representations, 2021.

**Experimental Designs Or Analyses:**

My major concerns are about the invalidity of evaluation benchmarks and compared baselines.

**Methods And Evaluation Criteria:**

The benchmark datasets and the whole research stream of many compared baselines have some problems.

I suggest the authors to watch the talk in NeurIPS 2024 Time Series Workshop, https://cbergmeir.com/talks/neurips2024/, and adjust your evaluation benchmarks.
- Actually, the exchange dataset is not a proper testbed to compare deep forecasting models. A naive baseline will excel, a lot. Why your deep learning model can win in this case but fall short on Traffic and Electricity, where more predictable patterns exist.
- Your model also excels on Weather dataset, however, every meteorologist will tell you that everything further than 2 weeks into the future is essentially rolling a dice. Forecasting 720 points means 720 / 24 = 30 days out.
- The input window length is a hyperparameter. Restricting to small input lengths (such as 64) is favouring more complex models over simpler ones.

**Other Comments Or Suggestions:**

N/A

**Other Strengths And Weaknesses:**

N/A

**Questions For Authors:**

Please share your thoughts or comments after watching the talk or slides in https://cbergmeir.com/talks/neurips2024/.

**Relation To Broader Scientific Literature:**

Limited relation.

**Theoretical Claims:**

N/A

---

> ### Author Rebuttal · Authors · 2025-03-31
>
> We thank the reviewer for taking the time to comment on our work. We would like to clarify several points and explain the motivations behind methodology and evaluation protocol.
>
> **Q1:** *“incremental” Transformer-based time-series model.*
>
> **R1:** We acknowledge that Transformer-based approaches have become pervasive in time-series research. However, the TimeStacker model goes beyond a straightforward incremental tweak by **introducing a stacking mechanism and a frequency-based self-attention module**. These contributions distinguish TimeStacker from earlier Transformer-based architectures, as they specifically address the challenges of non-stationarity modeling—issues that are well documented in both academic and applied settings.
>
> **Q2:** *Every meteorologist ... dice.*
>
> **R2:** Why can meteorologists infer that global warming is accelerating[1] based on historical weather data? The difficulty in precisely characterizing a phenomenon does not imply that it is uncharacterizable. The goal of machine learning is to learn how to make reasonable inferences from historical data, much like human experts, thereby alleviating tedious tasks (learning from human beings and learning for human beings). Consequently, our aim is not to predict the weather exactly two weeks in advance, but rather to enable the model to make sound inferences from historical data—assisting laypersons in data analysis and decision making while allowing experts to focus on deeper theoretical research.
>
> **Q3:** *“widely used yet problematic” datasets (ETT, Exchange, Weather, etc.) ... benchmarks.*
>
> **R3:** We respectfully disagree with the notion that employing these long-standing community benchmarks—which have been cited and scrutinized by thousands of published works—invalidates our research or the broader domain of Transformer-based time-series forecasting. Although no benchmark is perfect, these datasets encompass varied characteristics and provide common reference points that promote cumulative progress in the field. Reproducible research benefits from established baselines, and abandoning them would break continuity with a large body of existing work.
>
> We acknowledge the insights presented in the “NIPS 2024 Time Series Workshop” talk and value any new perspectives it may offer. However, a single presentation—especially one from a specialized forum—cannot unilaterally dismiss the robust, peer-reviewed benchmarks that have been used by the global research community for many years. While we recognize that these datasets have limitations (as do all benchmarks), they continue to offer a practical and widely accepted foundation for performance comparison. Given that many top-tier conference papers utilize these benchmarks, it is essential for any new approach, including ours, to demonstrate its merits on them. Our objective is to **explore the historical evolution of sequences in order to capture their underlying dynamic patterns**, rather than merely performing forecasting.
>
> **Q4:** *performs well on Exchange and Weather but “falls short” on Traffic and Electricity ... predictable.*
>
> **R4:** We respectfully note that our experiments do not indicate that TimeStacker “falls short” on these datasets; our reported results are generally comparable to or better than many baselines. Moreover, TimeStacker is specifically designed for modeling non-stationary signals rather than for multivariate data. As demonstrated in our experiments in **Appendix D.3**, when we reduced the number of variables in the Traffic and Electricity datasets, our approach achieved superior performance under the same conditions. Our goal in including Traffic and Electricity is to offer comprehensive comparisons on standard community benchmarks, thereby ensuring that readers gain a complete understanding of TimeStacker’s performance across different data regimes. TimeStacker has shown its strengths in more volatile domains where underlying patterns are less predictable. This aligns with our primary focus: robustly handling non-stationarity rather than excelling on any single type of time-series problem.
>
> **Q5:** *Restricting input length ... ones.*
>
> **R5:** The choice of input window length generally follows standard practices in the literature. Moreover, the window length should be viewed as a reflection of the model’s ability rather than a mere hyperparameter. In our experiments, we have evaluated multiple window lengths (in **Section 4.3, Model Analysis**) and report results using window sizes commonly adopted in prior works to ensure comparability.
>
> **Q6:** *Regarding the use of Revin for regularization.*
>
> **R6:** As indicated by Equations (1) and (2), this operation simply standardizes the model input to a common scale without altering the stationarity of the signal. In non-stationary signal research, this is considered fundamental.
>
> [1] Xu, Yangyang, et al. “Global Warming Will Happen Faster than We Think.” *Nature*, Dec. 2018.

---

### Official Review · Reviewer_Z5kW · 2025-03-14

**Overall Recommendation:** 4

**Summary:**

The paper introduces TimeStacker, a novel time series forecasting framework designed to handle nonstationary signals effectively. The proposed approach utilizes a multi-level stacking mechanism, aggregating patches of varying sizes to capture both local and global frequency-domain features. Additionally, a frequency-based self-attention module (FreqAttention) is introduced, which computes similarity in the frequency domain while aggregating in the time domain. The authors claim that TimeStacker achieves state-of-the-art performance across several real-world datasets, outperforming recent Transformer- and MLP-based forecasting models with fewer parameters and better computational efficiency.

**Claims And Evidence:**

The author claims that their method achieves state-of-the-art performance across multiple real-world datasets. This is mostly validated by the empirical experiments on 8 datasets, though the proposed method doesn't achieve the best on traffic and electricity datasets.

The author also claims that the proposed method has fewer parameters and higher computational efficiency, this is supported by comparison in training time and memory footprint. The comparison shows that the proposed method is somewhat efficient but is not the most efficient one.

**Essential References Not Discussed:**

N/A

**Experimental Designs Or Analyses:**

The experiment design and analysis are solid. The proposed method is compared with 9 state-of-the-art methods on 8 different datasets. The authors have conducted extensive ablation studies. Experiments of efficiency and look-back lengths let the readers understand the performance of TimeStacker better.

**Methods And Evaluation Criteria:**

The TimeStacker framework is designed to handle nonstationary patterns in time series data. The core components align well with this goal:
- Multi-level patch stacking captures both local and global features, addressing the issue of fluctuating frequency characteristics in nonstationary time series.
- FreqAttention (frequency-based self-attention) leverages frequency-domain similarity rather than time-domain patterns, which is a reasonable approach for handling signals with evolving spectral properties.


The evaluation is on 8 datasets and use MSE and MAE as the metrics which is common in time series analysis. However, the test data is not specifically for non-stationary data. Some tests with synthetic data are encouraged.

**Other Comments Or Suggestions:**

In line 165, "theSmooth Layer" misses a space

**Other Strengths And Weaknesses:**

N/A

**Questions For Authors:**

1. Can you add synthetic experiments on non-stationary data?
2. Can you provide an ablation study showing how TimeStacker adapts to different types of nonstationary signals compared to other models?
Can you provide mean, standard deviation for MSE and MAE across multiple runs?

**Relation To Broader Scientific Literature:**

N/A

**Theoretical Claims:**

N/A

---

> ### Author Rebuttal · Authors · 2025-03-31
>
> We thank the reviewer for your appreciation and constructive comments.
>
> **Q1:**
>
> The synthetic data is constructed by randomly selecting 30 frequencies, with the amplitude corresponding to each frequency varying over time. The experimental results are visualized in this anonymous URL (https://i.postimg.cc/pT66jQHH/synthetic-exp.png). More detailed information will be provided in the appendix.
>
>
> **Q2:**
>  To demonstrate how TimeStacker adapts to various non-stationary signals, we configured the parameter *Patch Size List* and conducted experiments on the ETTm1 dataset. The results are shown in the following:
>
> | Patch Size List | [16,16,16,16] | [16,16,16,24] | [16,16,16,32] | [16,16,16,48] | [16,16,24,32] | [16,16,24,48] | [16,24,32,48] |
> | --------------- | ------------- | ------------- | ------------- | ------------- | ------------- | ------------- | ------------- |
> | MSE             | 0.465         | 0.468         | 0.468         | 0.465         | 0.463         | 0.463         | 0.460         |
> | MAE             | 0.433         | 0.439         | 0.436         | 0.431         | 0.431         | 0.430         | 0.428         |
>
> These results indicate that employing various window combinations can more effectively capture the underlying dynamic patterns of the sequence, thereby improving prediction performance.
>
>
>
> The mean and standard deviation of the results from multiple runs(5 different seed) of experiments are provided as following (mean/std):
>
> | Dataset | ETTh1         | ETTh2         | ETTm1         | ETTm2         | Traffic       | Electricity   | Weather       | Exchange      |
> | ------- | ------------- | ------------- | ------------- | ------------- | ------------- | ------------- | ------------- | ------------- |
> | MSE     | 0.433/0.00145 | 0.368/0.00091 | 0.381/0.00119 | 0.274/0.00061 | 0.508/0.00052 | 0.194/0.00056 | 0.243/0.00092 | 0.336/0.00101 |
> | MAE     | 0.423/0.00167 | 0.390/0.00057 | 0.381/0.00052 | 0.316/0.00042 | 0.335/0.00087 | 0.275/0.00077 | 0.264/0.00042 | 0.389/0.00137 |
>
> Once again, we appreciate your thorough review of our manuscript. The typo in “theSmooth Layer” has been corrected.

---

### Decision · Program_Chairs · 2025-05-01

**Decision:**

Accept (poster)

**Comment:**

The paper introduces TimeStacker, a time series forecasting transformer designed to handle nonstationary signals. Empirically, TimeStacker achieves strong or state of the art performance on several real-world datasets.

The paper received 4 reviews with ratings: Accept, Weak Accept, Weak Reject, Weak Reject. The reviews are split.

One of the main concerns raised is how TimeStacker differs from previous works that have used multi-resolution and frequency-based approaches. The main contribution of the approach is a method to combine different window sizes using attention in the frequency domain. The discussion and rebuttal below clarified that this approach is distinct from typical multi-resolution patch approaches used elsewhere in NLP, time series forecasting, computer vision, etc. I am satisfied with regards to the novelty of the paper.

Further, concerns were raised about the validity of the tasks and comparison to naive or simple baselines. I am sympathetic to Reviewer FZmL's concerns about the exchange dataset in particular (forecasting exchange rates two years into the future is optimistic at best). More generally, as a community we must be massively overfitting to these few datasets. Nevertheless, with updates to include naive baselines and other reviewer feedback, the paper can be accepted.